# Structural insight into Okazaki fragment maturation mediated by PCNA-bound FEN1 and RNaseH2

Yuhui Tian [ID][1], Ningning Li [ID][1,2], Qing Li [ID][3] & Ning Gao [ID][1,2,4,5][✉]

## Abstract

PCNA is a master coordinator of many DNA-metabolic events. During DNA replication, the maturation of Okazaki fragments involves at least four DNA enzymes, all of which contain PCNA-interacting motifs. However, the temporal relationships and functional modulations between these PCNA-binding proteins are unclear. Here, we developed a strategy to purify endogenous PCNA-containing complexes from native chromatin, and characterized their structures using cryo-EM. Two structurally resolved classes (PCNA-FEN1 and PCNA-FEN1-RNaseH2 complexes) have captured a series of 3D snapshots for the primer-removal steps of Okazaki fragment maturation. These structures show that product release from FEN1 is a rate-liming step. Furthermore, both FEN1 and RNaseH2 undergo continuous conformational changes on PCNA that result in constant fluctuations in the bending angle of substrate DNA at the nick site, implying that these enzymes could regulate each other through conformational modulation of the bound DNA. The structures of the PCNA-FEN1-RNaseH2 complex confirm the toolbelt function of PCNA and suggests a potential unrecognized role of RNaseH2, as a dsDNA binding protein, in promoting the 5′-flap cleaving activity of FEN1.

**Keywords** PCNA; FEN1; RNase H2; Okazaki Fragment; Cryo-EM
**Subject Categories** DNA Replication, Recombination & Repair; Structural Biology

## Introduction

DNA replication is a conserved process across species, following the same principle of semi-conservative replication (O'Donnell et al, 2013). The leading strand is synthesized continuously in the 5′ to 3′ direction, while the lagging strand is synthesized discontinuously in short fragments called Okazaki fragments. DNA primase-polymerase alpha complex synthesizes the initial RNA-DNA primer for both strands. Subsequently, DNA Pol δ (polymerase delta) and Pol ε (polymerase epsilon) are responsible for the elongation of the lagging and leading strands, respectively (Garg and Burgers, 2005; O'Donnell et al, 2013). The proper function of these DNA polymerases depends on tens of accessory proteins. Among them, PCNA (Proliferating Cell Nuclear Antigen) binds to the polymerases and acts as a sliding clamp to tether them on DNA, enhancing their processivity (Prelich et al, 1987; Tan et al, 1986). Eukaryotic PCNA is a homotrimer, forming a ring encircling double-stranded DNA (Kelman, 1997). The interactions between PCNA and these polymerases are mediated by the so-called PCNA-interacting protein-boxes (PIP box) within different regions of their sequences. Besides DNA polymerases, over 200 proteins involved in various DNA metabolic processes, including DNA replication, repair, epigenetic regulation, and cell cycle regulation, also harbor PIP boxes or PIP-like motifs. The consensus sequence is QXXhXXaa where h is one of the I, L, V, or M residues, a is one of the F, W, or Y, and X can be any residue (Choe and Moldovan, 2017). Each PCNA monomer contains a PIP box docking site, and a PCNA ring has the potential to recruit up to three PIP box-containing proteins at a time. Therefore, PCNA has been proposed to be a master coordinator of different DNA metabolic processes (Choe and Moldovan, 2017; Kelman, 1997).

One example of the PCNA-coordinated process is the lagging strand maturation, which involves at least four PCNA-interacting enzymes, Pol δ, RNaseH2 (Ribonuclease H2), FEN1 (Flap endonuclease 1), and LIG1 (DNA ligase 1) (Sun et al, 2023). The removal of the RNA-DNA primer requires the coordination of RNaseH2 and FEN1 (Zheng and Shen, 2011); but this primer digestion process in vivo is not fully clear, and may be carried out in a network of pathways with parallel routes. A major route involves Pol δ and FEN1 as main players. In this model, after Pol δ starts strand displacement to invade into the DNA-RNA primer region of the previous Okazaki fragment, a short 5′-flap (2 to 10 nt) is generated, which is then removed by FEN1 (Sun et al, 2023). The primer is usually ~30 nt in length, including 8–12 nt of RNA and 10–20 nt of DNA (Baranovskiy et al, 2016; Sun et al, 2023). This displacement-digestion process could repeat (Maga et al, 2001; Rossi and Bambara, 2006), in a mechanism called nick translation, until the primer is fully removed. PCNA is essential for the processivity of Pol δ in both Okazaki fragment synthesis and RNA-DNA primer strand displacement (Maga et al, 2001; Prelich et al, 1987; Tan et al, 1986), and could increase the in vitro nuclease activity of FEN1 by 10–50 fold (Li et al, 1995). In contrast to the

[1]State Key Laboratory of Membrane Biology, Peking-Tsinghua Joint Center for Life Sciences, School of Life Sciences, Peking University, Beijing, China. [2]Changping Laboratory, Beijing, China. [3]State Key Laboratory of Protein and Plant Gene Research, School of Life Sciences and Peking-Tsinghua Center for Life Sciences, Peking University, Beijing, China. [4]National Biomedical Imaging Center, Peking University, Beijing, China. [5]Beijing Advanced Center of RNA Biology (BEACON), Peking University, Beijing, China. [✉]E-mail: gaon@pku.edu.cn

major route, the minor pathway involves RNaseH2 additionally to pre-remove the RNA primer. RNaseH2 digests RNA residues in the RNA/DNA hybrid of the primer, only leaving a single ribonucleotide at the 5′-end (Cerritelli and Crouch, 2009). Pol δ fills the gap and continues the strand displacement. FEN1 takes turn to remove the last RNA residue and the rest DNA residues in the primer (Balakrishnan and Bambara, 2013) in two or more rounds of nick translation. RNaseH2 is a heterotrimer composed of RNaseH2A, H2B and H2C in human, and the PIP box of RNaseH2B is suggested to be responsible for its association with PCNA (Bubeck et al, 2011; Chon et al, 2009). The last step of the Okazaki fragment maturation is the nick filling by LIG1 (Ellenberger and Tomkinson, 2008), which contains two separate PIP boxes (Blair et al, 2022; Pascal et al, 2006).

Overall, during the Okazaki fragment maturation, PCNA interacts with all of the major factors, and together they contain more than five PIP boxes. Do these factors act in a strictly sequential manner? How does PCNA select and coordinate these factors in a productive way? Is there reciprocal allosteric regulation between these factors while attached to PCNA? All of these questions remain unanswered. As suggested by the in vitro assembled complexes of PCNA-Pol δ-FEN1 (Lancey et al, 2020) and PCNA-FEN1-LIG1 (Blair et al, 2022), one hypothesis is that PCNA acts as a toolbelt to keep at hand necessary factors required for the lagging strand processing to increase the overall efficiency.

In this work, we used PCNA as the bait protein and obtained PCNA-containing complexes from native chromatin of cultured HEK293 cells. Compositional and structural analyses confirm that PCNA is indeed a toolbelt in vivo, allowing the functional interplay between FEN1 and RNaseH2 during the lagging strand processing. Importantly, our data imply an unrecognized functional coupling between RNaseH2 and FEN1, in which RNaseH2 acts as a dsDNA binding protein upstream the nicked site, independent its catalytic activity, to maintain a specific configuration of the substrate DNA required for the FEN1-mediate 5′-flap removal. This finding exemplifies that PCNA binding proteins are more than separate tools in the PCNA toolbelt, and the simultaneous occupation of the substrate DNA by multiple factors could be employed to augment certain catalytic events.

## Results

### Purification of the endogenous PCNA-containing complexes from native chromatin

To obtain chromatin-associated PCNA-containing complexes, the chromatin fractions of HEK293 cells expressing N-terminally twin-strep tagged PCNA were first isolated following a standard protocol (Khan et al, 2020) (Fig. 1A). Although the overexpression level of PCNA was high through transient plasmid transfection, we found that twin-strep-PCNA were efficiently incorporated into the chromatin fractions (Fig. 1B; Appendix Fig. S1). Preliminary test of three chromatin fragmentation methods (MNase, benzonase digestion, and sonication) suggested that limited MNase digestion outperformed the other two in preserving nucleosome structures. Subsequently, solubilized supernatants from chromatin samples treated by MNase were incubated with anti-strep affinity beads. The eluates were subjected to one round of glycerol density gradient

centrifugation, and PCNA-containing complexes of different molecular weights were separated (Fig. 1B,C).

Negative staining electron microscopy (nsEM) and mass spectrometry were employed to characterize the samples in glycerol fractions. As expected, the size of particles increases with the fraction number. In the peak fractions of PCNA (Fractions 3–5), typical PCNA rings could be easily detected. Towards heavier fractions, particles are getting larger but the heterogeneity is also more obvious. In the Coomassie Brilliant Blue-stained protein gel, co-purified proteins display distinct patterns in different fractions. For example, FEN1 co-migrates with PCNA in most of the fractions (starting from Fraction 4), and in the following heavier fractions (starting from Fraction 6), more factors, including DNMT1, MSH2, MSH3, MSH6, RNaseH2 subunits and four histone proteins, started to appear at different fractions. In even heavier fractions, DDB1, subunits of CAF1 heterotrimer, RFC complex and MCM complex are clearly discernible on the gel. Among these PCNA-associated factors, DNMT1, MSH6, MSH3, CAF1A, FEN1 and RNaseH2B are known to interact with PCNA directly through their PIP motifs (Bubeck et al, 2011; Kleczkowska et al, 2001; Leonhardt et al, 1992; Li et al, 1995; Majka and Burgers, 2004). MSH2 can interact with MSH3 to form Mutα complex and form Mutβ with MSH6 to recognize mismatched bases (Pecina-Slaus et al, 2020), and the RFC complex is responsible for loading PCNA onto DNA (Majka and Burgers, 2004). In addition to these dominant bands on the gel, mass spectrometry analysis of the samples indicates a high enrichment of various proteins involved in PCNA-related DNA metabolic events (Fig. EV1A,B).

Therefore, these compositional analyses indicate that our sample preparation is valid in capturing highly specific PCNA-containing complexes. Together with the fact that four histones were among the most abundant proteins in the sample, it shows that we have successfully obtained endogenous PCNA-associated complexes involved in different DNA metabolic events in the context of nucleosomes, including DNA replication, Okazaki fragment processing, nucleosome assembly, mismatch repair and DNA methylation.

### Structural profiling of endogenous PCNA-containing complexes

Guided by the nsEM results (Fig. 1C), we selected Fractions 9–12 and 17–20 for cryo-EM sample preparation and data collection (Fig. 2A; Appendix Fig. S2A,B). Different strategies of particle-picking, two-dimensional (2D) and three-dimensional (3D) classification were tested.

Processing of the datasets from Fractions 9–12 led to the discovery of four representative classes, which are also evident from the 2D classification results (Fig. 2B–F). Three of them contain a characteristic PCNA ring, but also with different features. Subsequently, the particles were divided into groups for further 3D classification and refinement (Fig. 2G–J). The first class, which accounts for 47% particles, is a PCNA-FEN1 complex with a short DNA in the central channel of PCNA. The second one (6% particles) contains three subunits of RNaseH2, forming a stable PCNA-FEN1-RNaseH2 complex. The third class is dominated by nucleosome signals on both the 2D and 3D levels, although fuzzy densities could also be observed round the nucleosome. When the density map of this class is displayed at low threshold, extra non-

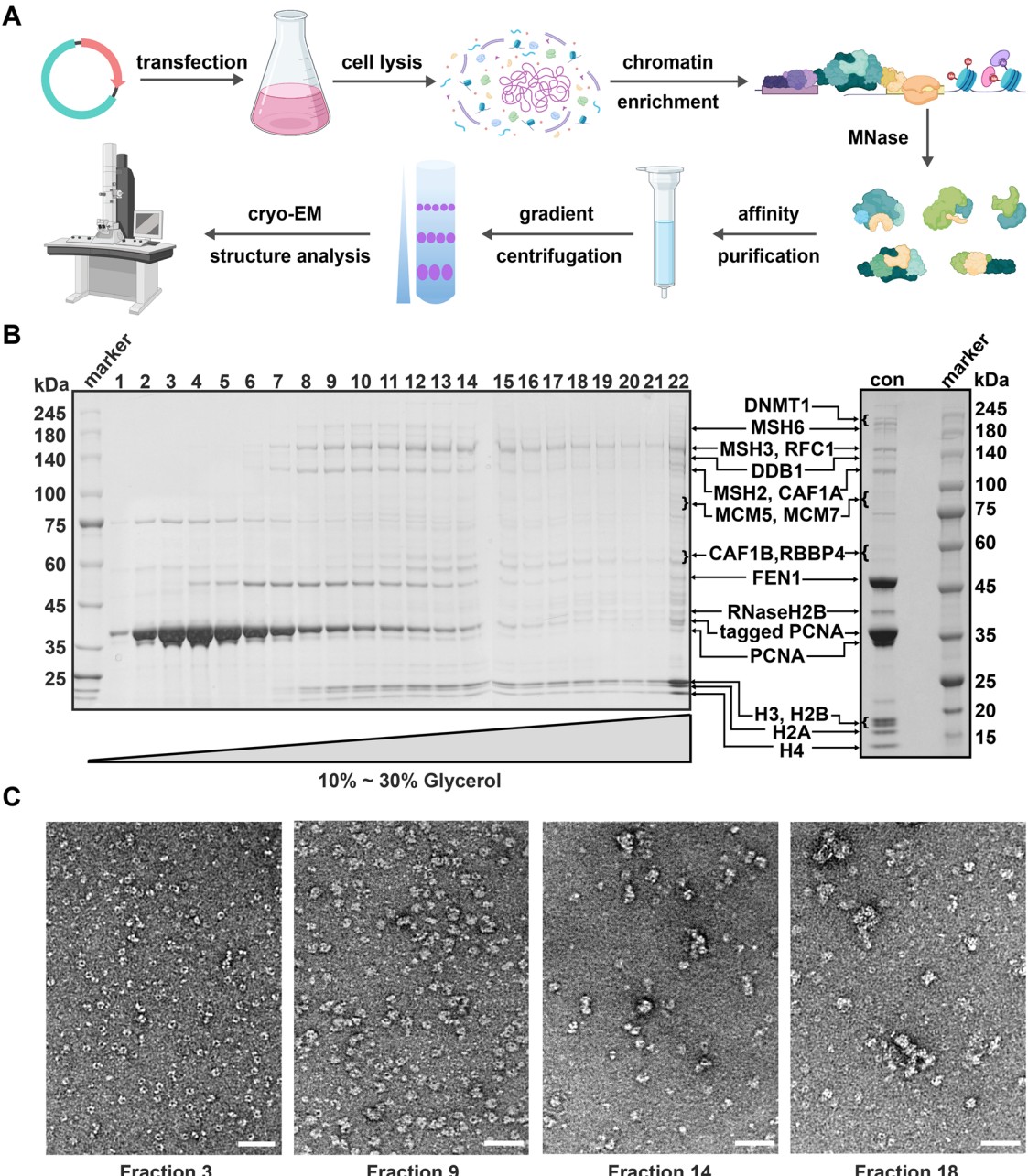

**Figure 1. Purification of endogenous PCNA-containing complexes from native chromatin.**

(A) Flowchart of the method for obtaining endogenous PCNA-containing complexes from native chromatin. *PCNA* plasmids with an affinity tag are transiently transfected into HEK293 cells. Cultured cells are collected and lysed with mild detergent Triton X-100 to obtain chromatin fraction. Then MNase is used to digest the chromatin to obtain soluble components, which are further purified through affinity purification and glycerol density gradient centrifugation. Complexes with different molecular weights are obtained, and the structures of the complexes are analyzed by cryo-EM. (B) Protein electrophoresis of the PCNA-containing endogenous complexes after glycerol density gradient centrifugation stained with Coomassie Brilliant Blue. Major bands on the gel were identified by mass spectrometry, and labeled. The "con" sample was a concentrated sample from fractions 10–12, which was subjected to MS analysis. (C) Samples of different fractions examined by negative staining (with uranium acetate). The scale bar in the panels is 50 nm. Source data are available online for this figure.

nucleosomal densities could be found on one face of the nucleosome. This nucleosome class is a minor population and further 3D classification failed to reveal the identity of unknown nucleosome-binding proteins. Very interestingly, the fourth class features a two-ringed structure, and each of them matches well with a PCNA ring. The dimer structure was determined at 8.7 Å

resolution (Fig. 2J). Since the physiological relevance of this dimer is not clear, it was not included in the following analysis. In contrast, image processing of the particles from Fractions 17–20 reveals few classes with discernible feature of PCNA. Some particles contain long DNA with additional protein densities (Appendix Fig. S2A,B). After 2D classification, a common feature of many

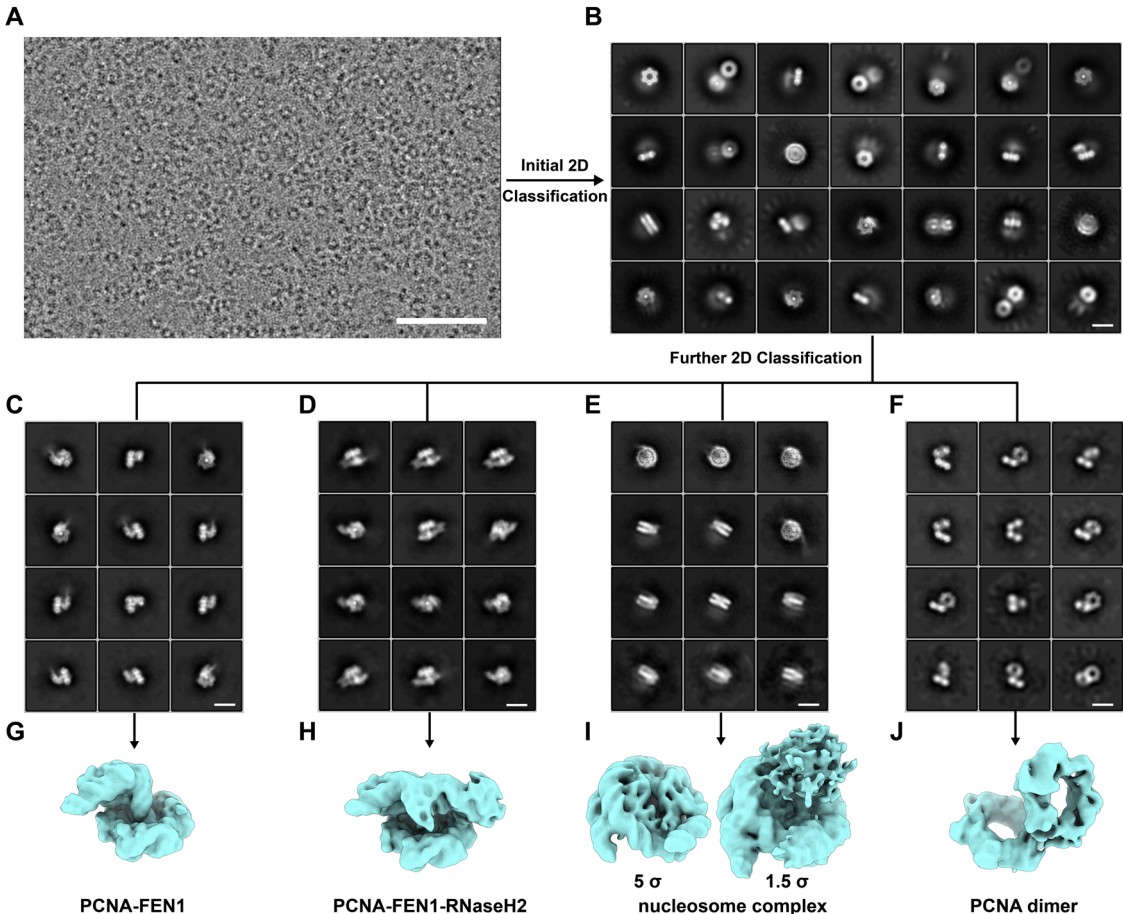

**Figure 2. Structural profiling of endogenous PCNA-containing complexes.**

(A) Representative micrography of the frozen samples prepared with Fraction 9–12 under cryo-EM. The scale bar in (A) is 50 nm. (B) Initial two-dimensional (2D) classification of all the picked particles. The scale bar in (B) is 10 nm. (C–F) 2D classification of four representative groups of particles with different features. These four classes of particles are PCNA-FEN1 (C), PCNA-FEN1-RNaseH2 (D), nucleosome complex (E), and PCNA dimer (F). The scale bar in (C–F) is 10 nm. (G–J) Three-dimensional (3D) reconstructions of PCNA-FEN1 (G), PCNA-FEN1-RNaseH2 (H), nucleosome complex (I), and PCNA dimer (J). The structures shown in (G–J) are in the same scale. The map of the nucleosome complex is shown at two different contour levels. Source data are available online for this figure.

classes is the nucleosome (Appendix Fig. S2C). Since PCNA is still abundant in these complexes of high-molecular weight, it suggests that at least some of these particles are PCNA-associated chromatin fragments.

## Cryo-EM structures of the PCNA-FEN1 complex

After classification and refinement of the PCNA-FEN1 particles, we obtained a series of cryo-EM maps with resolutions ranging from 3.5 to 3.8 Å (Fig. 3A–C; Appendix Figs. S3 and S4). Although the crystal structures of the FEN1-DNA complex and the PCNA complex with the PIP box of FEN1 have been previously determined (Bruning and Shamoo, 2004; Chapados et al, 2004; Tsutakawa et al, 2011; Tsutakawa et al, 2017), and low-resolution cryo-EM structure of the in vitro assembled PCNA-FEN1 complex has also been preliminarily characterized (Blair et al, 2022), this represents the first instance of obtaining high-resolution structures of the endogenous PCNA-FEN1 complexes.

In these structures, FEN1 binds to one monomer of the PCNA ring, and the other two PIP box accommodation sites are vacant

(Fig. 3A–C). FEN1 mainly relies on its PIP box to associate with PCNA. Similar to the previous crystal structure (Bruning and Shamoo, 2004; Chapados et al, 2004; Sakurai et al, 2005), the PIP box of FEN1, containing a short $3_{10}$-helix, is located in a hydrophobic pocket in the C-terminal domain of PCNA. The very C-terminus of FEN1, downstream of the PIP box, establishes an antiparallel β-sheet interaction with the IDCL (interdomain-connecting loop) of PCNA (Fig. 3D). A L-shaped DNA was found in all the structures. One strand of the DNA, which is the template strand, is continuous, and the daughter strand is disconnected at the catalytic center of FEN1 (Fig. 3C,E,F), resulting in a ~90° bending of the bound DNA. The two terminal regions of the bound DNA are in double-stranded form. While the downstream FEN1-bound dsDNA is about 12 bp, the upstream dsDNA varies in length (14–19 bp in different structures) and passes through the PCNA ring (Figs. 3E and EV2A–D). The 5′-flap of the daughter strand threads through FEN1, and except the three proximal nucleotide residues, the rest of the 5′-flap is untraceable (Fig. 3B,E). The helical clamp above the active center of FEN1 is well folded and interacts with these three residues to stabilize them. At the 3′-end of the

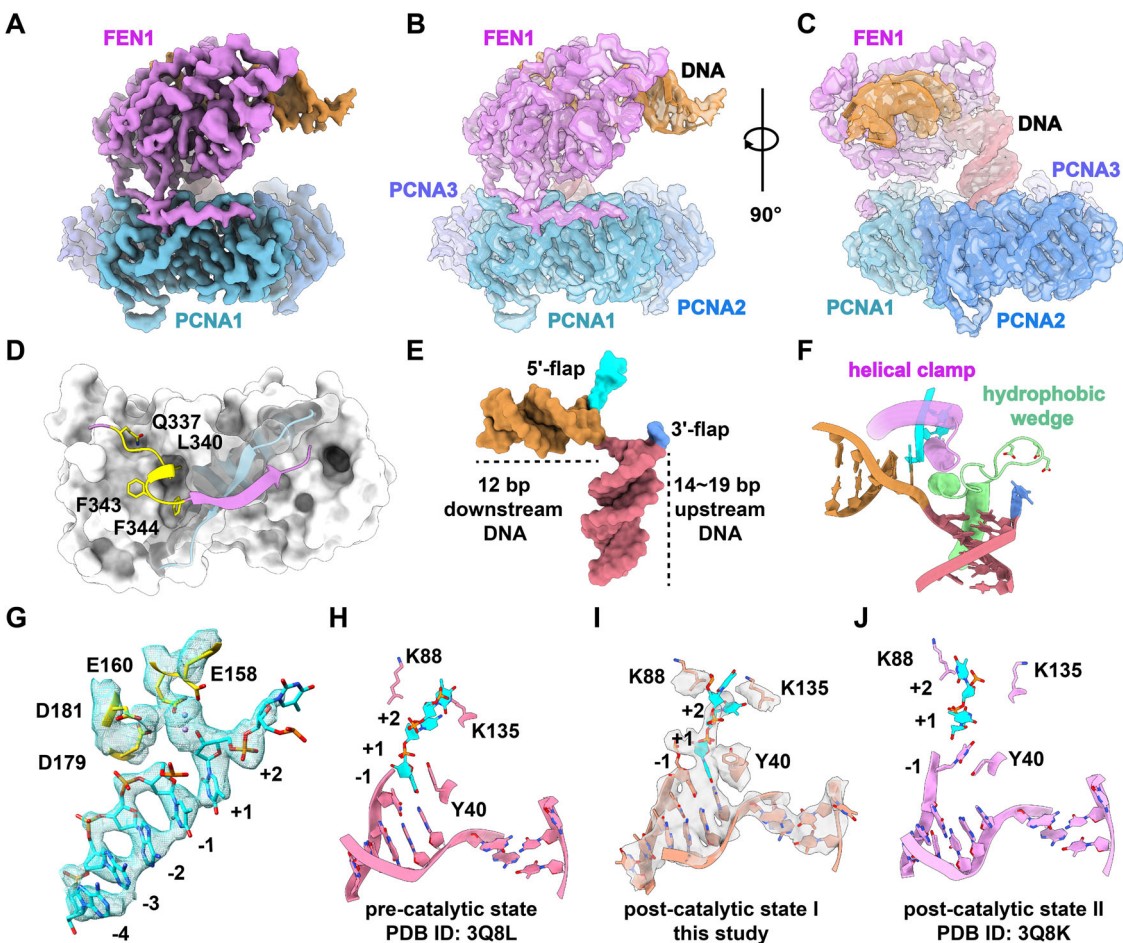

**Figure 3. Cryo-EM structure of the PCNA-FEN1 complex.**

(A) The density map of the PCNA-FEN1 complex. (B) Atomic model of the PCNA-FEN1 complex fitted into the density map displayed with the same orientation as (A). (C) Atomic model of the PCNA-FEN1 complex fitted into the density map displayed with orientation rotated 90° along the y-axis relative to (B). (D) PCNA-interacting protein-box (PIP box) and the C-terminus of FEN1 bound to PCNA. The PIP box and the C-terminus of FEN1 are colored yellow and magenta, respectively. The IDCL of PCNA is colored light blue. (E) The endogenous DNA of PCNA-FEN1 complex. The 5′-flap is painted in cyan and the 3′-flap in blue. The downstream double-stranded region is 12 bp and the upstream DNA varies in length (14–19 bp in different structures). (F) Zoomed-in view of 5′-flap and 3′-flap binding pocket. The helical clamp of FEN1 is pained in purple, and the hydrophobic wedge in spring green. The side chains of the three negatively charged amino acids E56, E57, and E59 in hydrophobic are shown which ensure that FEN1 only binds to the one nucleotide flap of the 3′-end. (G) The density map near the cleaved 5′-flap. The negatively charged amino acids E158, E160, D179, and D181 in the active center (colored yellow) participate in the coordination of metal ions. There is a break in the density between the nucleotide at the +1 position and the nucleotide at the −1 position. (H–J) Zoomed-in view of different states of FEN1 and DNA. The 5′-flap is colored cyan, and the remaining DNA of the pre-catalytic state (H) (PDB ID: 3Q8L) (Tsutakawa et al, 2011) is colored moldy pink. The structure obtained in this study (I) is painted in orange. Although the 5′-flap is cleaved, the nucleotide at position +1 is still complementary to the template strand, and the position of Y40 has rotated relative to that in the pre-catalytic state. This state revealed in our study is named post-catalytic state I. The state after cleavage (PDB ID:3Q8K) (Tsutakawa et al, 2011) is post-catalytic state II (J), painted in light pink, in which the nucleotide base at position −1 forms a π–π stack with the benzene ring of Y40.

discontinued daughter strand, only one overhang residue is visible, which is stabilized by the hydrophobic wedge of FEN1 (Fig. 3C,F). The high-resolution density maps show that the phosphodiester bond of the 3′-end of the 5′-flap DNA in the catalytic center of FEN1 is broken (Fig. 3G; Appendix Fig. S4A), indicating that the DNA substrate is in the post-catalytic state.

Overall, FEN1 adopts nearly identical overall conformations as seen in previous FEN1-DNA crystal structures (Tsutakawa et al, 2011; Tsutakawa et al, 2017), with the RMSD being 0.89 Å. In the crystal structures of the FEN1-DNA complex in the pre-catalytic state prepared with catalytically deficient mutant of FEN1 (Tsutakawa et al, 2011), the single-stranded 5′-flap is intact, and

the nucleotide at +1 position pairs with the template strand (Fig. 3H). In contrast, in the crystal structure of the post-catalytic state (Tsutakawa et al, 2011), the 5′-flap moves away from the active center after the bond cleavage, and the nucleotide at +1 position is distant from the template strand. In addition, the nucleotide at −1 position is also unpaired to the template DNA strand (Fig. 3J). Interestingly, the DNA substrate in our structures is different from all previous ones: The cleaved 5′-flap remains stably bound in the enzymatic center, with the +1 nucleotide of the 5′-flap still basepairing with the template strand (Fig. 3I; Appendix Fig. S4A). These structural differences are closely related to the spatial positions of Y40, K88, and K135 in FEN1. For example, the

phenyl ring of Y40 of FEN1 stacks with the base of +1 nucleotide in pre-catalytic state, whereas in the post-catalytic state, it stacks with the base of −1 nucleotide (Fig. 3H,J). In our structures, this stacking interaction of Y40 with +1 nucleotide observed in the pre-catalytic state has been disrupted, due to a rotation of the phenyl ring (Fig. 3I). Therefore, our structures have captured an earliest state after bond cleavage. Together with previous structures, they represent a series of distinct, temporally related snapshots for the FEN1-mediated 5′-flap processing.

## Structural dynamics of the PCNA-FEN1 complex in the post-catalytic state

During the 3D classification, the PCNA-FEN1 particles were grouped in eight classes for structural refinement (Appendix Fig. S3). Two major differences on the bound DNA in these states are their orientations relative to the PCNA ring and the bending angles between the upstream and downstream dsDNA segments (Fig. EV2 and Movie EV1). The DNA was seen to have a rotation around the channel axis of PCNA (Fig. EV2A–D). These states are named state A to state H based on the clockwise position of the downstream DNA in the side view (Fig. EV2B). We also quantitatively compare the bending angle among these states (Appendix Fig. S5A). The angle between the upstream and downstream DNA in state A is 107°, and it increases to 129° in state H. By aligning PCNA in different states, the changes in DNA angles can be observed more intuitively (Fig. EV2B). Similar to the angle measurement, we also measured the displacements of the DNA ends in these structures (Appendix Fig. S5B). As a result, the displacement of the 5′-flap and the 3′-flap could be as large as 31 Å. Notably, although the position of upstream dsDNA segment relative to PCNA changes greatly in these different states, FEN1 moves together with the downstream fragment accordingly. For example, the maximal movement of the helical clamp of FEN1 could reach 26 Å (Fig. EV2E; Appendix Fig. S5C). The flexibility of FEN1 relative to PCNA relies on the flexible linker between the PIP box and the main body of FEN1. An interesting observation is that the C-terminal end of the connecting helix immediately upstream PIP box could switch from helical to loop conformations to accommodate the large change in the orientation of FEN1 (Fig. EV2F).

To explore whether the different states of the PCNA-FEN1 complex are discrete or continuous, we used Relion to do multibody refinement (Nakane et al, 2018) and cryo-SPARC to perform 3D variability analysis (Punjani and Fleet, 2021), respectively. FEN1 and PCNA were used as two rigid bodies. Multibody refinement results revealed six dominant eigenvectors (larger than 5% of the total variance). The first one contributes to over 20% of the overall variance (Appendix Fig. S5D and Movies EV2–3). However, the distribution of particles along the first eigenvector only displays a single peak, indicating that the changes between these states are continuous rather than discrete (Appendix Fig. S5E). Similarly, 3D variability analysis shows that the distributions of particles on different eigenvector planes are all single-peaked (Appendix Fig. S5F).

In conclusion, these analyses show that FEN1 is dynamic on the PCNA ring, which could undergo continuous conformational change to accommodate the DNA length variation between PCNA and the catalytic center of FEN1. This flexibility might be important for the in vivo function of FEN1, because it does not require a specific length on the upstream dsDNA for FEN1 to exert its catalytic function.

## Cryo-EM structures of the PCNA-FEN1-RNaseH2 complex

The population of the PCNA-FEN1-RNaseH2 particles is rather small, about 12% of the PCNA-FEN1 particles. After 3D classification, four structures of the PCNA-FEN1-RNaseH2 complex could be obtained at resolutions between 5 and 6 Å (Fig. 4A–C; Appendix Fig. S6). Pseudo-atomic models were built using the structures of three RNaseH2 subunits predicted by AlphaFold2 (Jumper et al, 2021; Varadi et al, 2022). The DNA in this ternary complex is generally similar to that of the PCNA-FEN1 complex, with a ~90° kink in the middle. The three subunits of RNaseH2 form an elongated shape, with the catalytic subunit RNaseH2A bound to the upstream dsDNA region. The other two subunits of RNaseH2 are free of DNA interaction. Interestingly, there is no direct interaction between FEN1 and RNaseH2, and the latter relies on PCNA and DNA to form a stable complex (Fig. 4A–C).

Compared with the crystal structure of human RNaseH2 (PDB 3PUF) (Reijns et al, 2011), our structure of RNaseH2 is more compact, with RNaseH2B and RNaseH2C moving closer to PCNA (Fig. 4D). Previously, based on a crystal structure of PCNA in complex with the PIP box of RNaseH2B, it is proposed that RNaseH2B is responsible for anchoring RNaseH2 to PCNA during Okazaki fragment processing (Bubeck et al, 2011). In contrast, in our endogenous PCNA-FEN1-RNaseH2 complex, the PCNA association is mediated by RNaseH2A, not RNase H2B (Fig. 4B). In the C-terminal region of RNaseH2A, a non-canonical PIP box (265-LRKITSYF-272) is identified, which locates to the conventional PIP box docking site of one PCNA monomer (Fig. 4E,F). However, when compared with the binding mode of the PIP box of FEN1, the intermediately downstream sequence of RNaseH2A PIP box does not form a β-sheet interaction with PCNA, as seen in the contact between PCNA and FEN1. Instead, after the $3_{10}$-helix at the end of the PIP box, the C-terminal sequence of RNaseH2A makes a U-turn, extends away from PCNA and eventually interacts with both RNaseH2C and RNaseH2B (Fig. 4A,B,E). The very C-terminal fragment of RNaseH2A ends at the interface between RNaseH2B and RNaseH2C.

Upon the structural alignment of the PCNA-FEN1-RNaseH2 complex with the PCNA-FEN1 complex using PCNA as reference, FEN1 and the downstream DNA in the ternary complex further deviate from the central axis of PCNA, and RNaseH2A would clash with FEN1 in the PCNA-FEN1 complex (Fig. 4G). This suggests that the binding of RNaseH2 to the PCNA-FEN1 complex would induce a significant spatial rearrangement of FEN1, with the helical clamp moving as much as 38 Å (Fig. 4G).

Altogether, our data provide direct evidence that PCNA is an in vivo platform for the simultaneous association of multiple factors. In the previous models (for example, see ref. (Sun et al, 2023)), these factors were depicted to act one by one to catalyze the maturation of the lagging strand. Our structure of the PCNA-FEN1-RNaseH2 complex show that the same DNA substrate could stably bind to two enzymes, suggesting the presence of certain kinetic inter-dependence for these two factors. Importantly, in all the ternary structures, the PIP box of RNaseH2 always associates

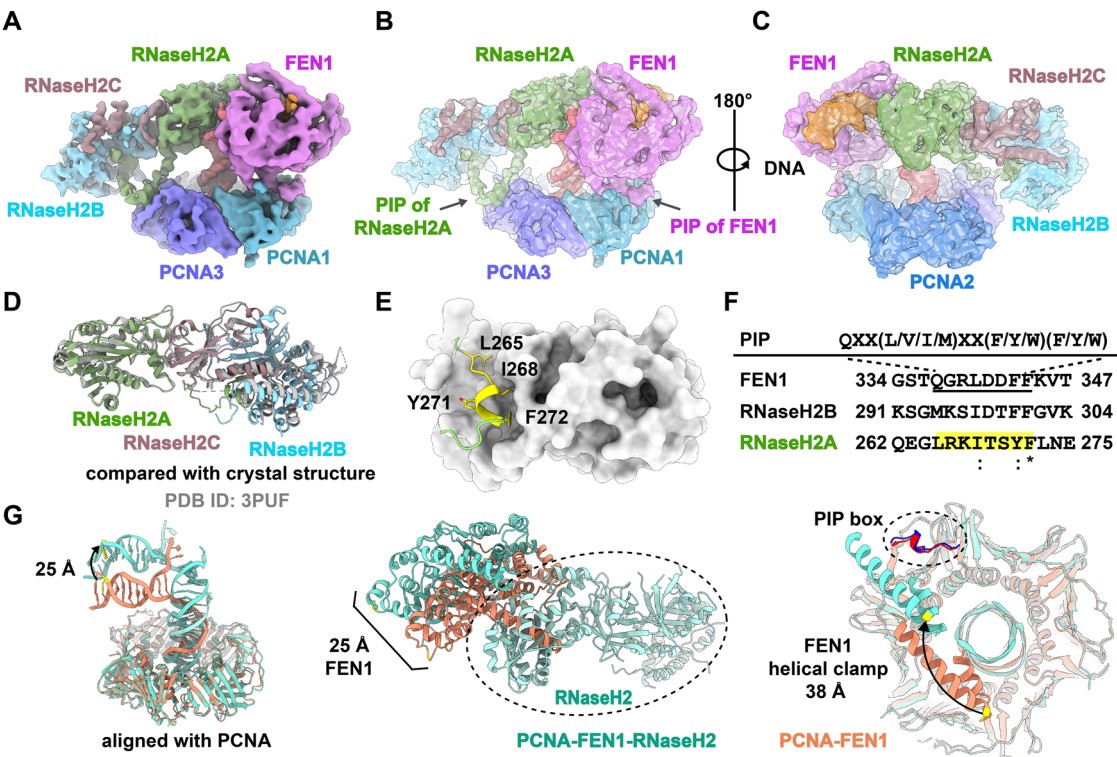

**Figure 4. Cryo-EM structure of the PCNA-FEN1-RNaseH2 complex.**

(A) The density map of the PCNA-FEN1-RNaseH2 complex. (B) Atomic model of the PCNA-FEN1-RNaseH2 complex fitted into the density map displayed with the same orientation as (A). (C) Atomic model of the PCNA-FEN1-RNaseH2 complex fitted into the density map displayed with orientation rotated 180° along the y-axis relative to (B). (D) Structural comparison of endogenous RNaseH2 with the reported crystal structure (PDB ID:3PUF) (Figiel et al, 2011). The crystal structure is colored gray. (E) PIP box of RNaseH2A bound to PCNA. The PIP box is colored yellow. (F) Sequence alignment of the PIP boxes of FEN1, RNaseH2B, and RNaseH2A. X represents any amino acid. (G) Structural comparison between the state D PCNA-FEN1 and PCNA-FEN1-RNaseH2 complexes, aligned with PCNA bound to FEN1 as reference. The Cα atoms of FEN1 A116 in the two complexes are 38 Å apart. In the right panel, the PIP box of FEN1 in the PCNA-FEN1 complex is colored red, and the PIP box of FEN1 in the PNA-FEN1-RNaseH2 complex is colored blue.

with the same PCNA monomer relative to the one occupied by FEN1, further indicating that the spatial relationship of two factors is deterministic and not random.

## PIP box of RNaseH2A binds to PCNA in vitro but is not involved in the initial recruitment to PCNA

In the structure of the endogenous PCNA-FEN1-RNaseH2 complex, we unexpectedly identified a PIP box in RNaseH2A, which binds to PCNA, rather than that of RNaseH2B as previously suggested (Bubeck et al, 2011). We then purified wild-type and four types of mutant RNaseH2 complexes, containing enzymatically dead mutant of RNaseH2A (IM_DD/AA), PIP-box mutant of RNaseH2A (PIPA_YF/AA), PIP-box mutant of RNaseH2B (PIPB_FF/AA), or double PIP-box mutant (with both RNaseH2A and RNaseH2B mutated), and incubated them with PCNA in vitro and tested their binding to PCNA using size-exclusion chromatography (Appendix Fig. S7). The IM_DD/AA mutant showed no defect in binding to PCNA. Single PIP mutants of RNaseH2 (PIPA or PIPB) could still bind to PCNA, but the peaks of mutant complexes migrated slightly slower, suggesting that their binding is less stable. In contrast, when both PIP boxes were mutated, RNaseH2 no longer interacted with PCNA. To examine the in vivo

function of RNaseH2A PIP box, we constructed a plasmid encoding all three subunits, GFP-RNaseH2A, mCherry-RNaseH2B and RNaseH2C, and transiently transformed the plasmids into U2OS cells. We found that wild-type RNaseH2A, RNaseH2B and PCNA co-localized at the replication sites (Fig. EV3).

However, when the RNaseH2B PIP box was mutated, RNaseH2A and RNaseH2B were evenly distributed in the nucleus and did not colocalize with PCNA. This result is consistent with previous work (Bubeck et al, 2011). In contrast, there is no defect in the PCNA colocalization when the PIP box of RNaseH2A was mutated. Given that the PIP boxes of RNaseH2A and RNaseH2B both binds to PCNA in vitro and in vivo, the function of these two PIP boxes might be different and stage-specific. The initial recruitment of RNaseH2 to the replication sites might mainly rely on the interaction between RNaseH2B PIP box and PCNA. Nevertheless, the exact role of these two PIP boxes requires further investigation.

## Structures of PCNA-DNA-FEN1-RNaseH2 complexes in different functional states

Similar to the PCNA-FEN1 complex, we found that the PCNA-FEN1-RNaseH2 complex also exists in multiple conformational

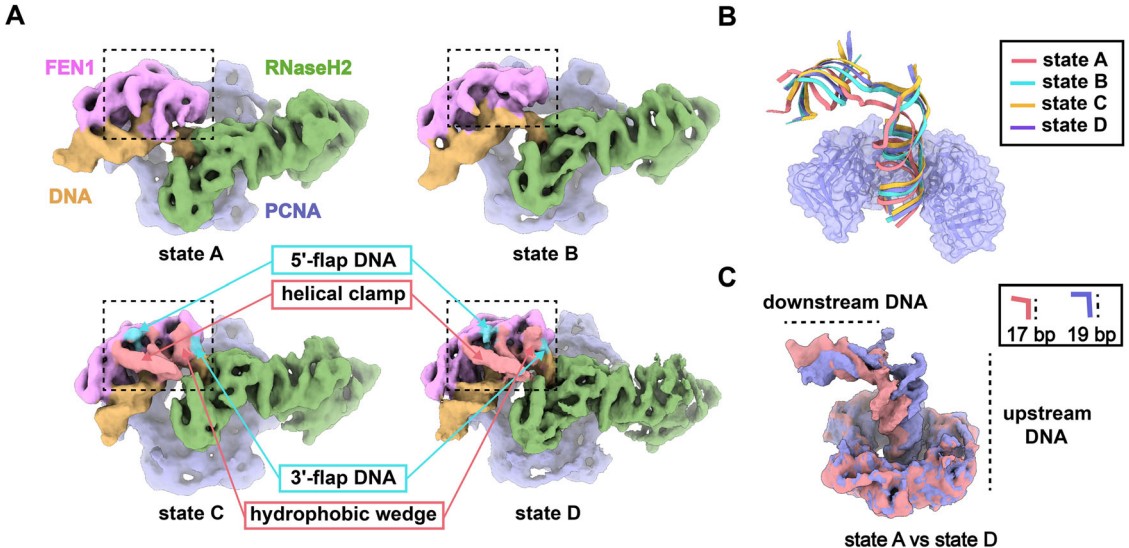

**Figure 5.   Different functional states of the PCNA-FEN1-RNaseH2 complex.**

(A) Cryo-EM density maps of the PCNA-FEN1-RNaseH2 complex in four different functional states. The helical clamp and hydrophobic wedge of FEN1 that bind to the DNA flaps have been stabilized in state C and state D, with the flap DNA clearly visible, whereas in state A and state B they are all relatively flexible. (B) Position and angle changes of DNA in different states with PCNA used as the reference of alignment. (C) Comparison of DNA in state A and state D. The DNA angle in state A is larger than in state D, while upstream DNA in state A is 17 bp, shorter than 19 bp in state D.

states (Fig. 5A; Appendix Fig. S6). The differences among these structures lie in the FEN1-DNA subcomplex (Fig. 5A–C, Movie EV4). In the maps of states A and B, the helical clamp and the hydrophobic wedge of FEN1, which are responsible for stabilizing the 5′-flap and the 3′-flap around the DNA kink, are disordered or partially disordered, and the density of the 5′-flap DNA is also absent. In state B, FEN1 and the downstream DNA together are seen to have a rotation relative to RNaseH2 (Fig. 5B), and the hydrophobic wedge of FEN1 have begun to be stabilized. In state C, the helical clamp and hydrophobic wedge have already formed, but the tip of the 5′-flap DNA is flexible. The structure of FEN1 in state D is very similar to FEN1 in the PCNA-FEN1 complex (Figs. 5A and 3C,F), with both the 5′- and 3′-flaps stabilized. These structural comparisons indicate that the four structures might have captured the dynamic changes of FEN1 during the 5′-flap processing: FEN1 and the 5′-flap DNA undergo conformational changes in a coordinated manner such that the 5′-flap is threaded through the enzyme active center of FEN1 (Movie EV4).

When PCNA is used as the reference for alignment, there are evident changes in the relative positions and orientations of FEN1 and RNaseH2 on the PCNA ring. Multibody refinement results show that there are nine major eigenvectors, of which the first two account for 35% of the total variance (Appendix Fig. S8A and Movies EV5–6). However, similar to what has been observed for the PCNA-FEN1 complex, the particle distributions along these eigenvectors are all unimodal, indicating that these states are also continuous snapshots, not discrete (Appendix Fig. S8B). 3D variability analysis of the PCNA-FEN1-RNaseH2 complex showed that particle clouds formed single clusters on eigenvectors planes, consistent with the results of multibody refinement (Appendix Fig. S8C).

## A possible second role of RNaseH2 as a dsDNA binding protein to promote FEN1-mediated 5′-flap removal

RNaseH2 and FEN1 are both endonucleases. RNaseH2 cleaves RNA within the RNA/DNA hybrid (Qiu et al, 1999), while FEN1 prefers to cleave ssDNA (Shen et al, 2005). The obtained structures show that RNaseH2 is still bound to the upstream double-stranded DNA. This observation is consistent with previous work showing that RNaseH2 strongly binds to the dsDNA in vitro (Coffin et al, 2011). We also showed that RNaseH2 could bind to the dsDNA of varying length, as short as 5 bp and PCNA significantly increased the dsDNA binding efficiency of RNaseH2 (Fig. EV4). In the current model of the lagging strand maturation, daughter strand elongation by Pol δ, RNA primer removal by RNaseH2, 5′-flap formation by strand displacement of Pol δ, 5′-flap cleavage by FEN1 and the nick filling by LIG1 are thought to occur sequentially (Sun et al, 2023). In our structures, the 5′-flap DNA has been generated, indicating that polymerase δ should have performed the strand displacement. However, RNaseH2, but not Pol δ remains bound to the double-stranded region of the upstream DNA. This double-stranded region should not contain any RNA residue, since the 5′-flap has already formed, which raises one interesting question why RNaseH2 is still in this region. In a recent in vitro study, the kinetics of the human Okazaki fragment maturation was measured in the presence or absence of different factors (Raducanu et al, 2022). One important finding was that the presence of RNaseH2 in the in vitro system could accelerate the overall rate of the Okazaki fragment maturation by 10 times to ~15 s (Raducanu et al, 2022). Although RNaseH2-catalyzed removal of the RNA primer should contribute to the overall maturation process, the time directly spent on the RNA removal (<3.9 s) is insignificant compared to the time of 150 s required to mature the entire

Okazaki fragment in the absence of RNaseH2 (Raducanu et al, 2022). Therefore, it appears that RNaseH2 might have an unrecognized, direct role in promoting the 5′-flap cleaving activity of FEN1. But how could this be achieved? The activity of FEN1 strictly depends on the configuration of its DNA substrate, requiring exactly one nucleotide flap on the 3′ upstream DNA (Chapados et al, 2004; Finger et al, 2009). Because the 5′-flap DNA is formed by strand displacement, there is an equal chance for this 5′-flap to invade the upstream dsDNA to generate a long 3′-flap (Fig. 6A–L). The presence of RNaseH2 in the double-stranded region of the upstream DNA would prevent the formation of the undesired long 3′-flap. In this way, it might promote the activity of FEN1, independent of its catalytic activity, by maintaining a specific configuration of the substrate DNA required for FEN1 action.

## Discussion

The structural analysis of the endogenous PCNA-containing complexes provided insights into the maturation steps of Okazaki fragment processing. In the PCNA-FEN1-RNaseH2 complex structures, FEN1 and DNA were observed to undergo coordinated conformational changes to thread the 5′-flap through FEN1. These conformational changes are consistent with the induced-fit mechanism proposed by in vitro experiments (Rashid et al, 2017) and support the model that FEN1 initially binds to the double-stranded DNA region before threading the 5′-flap through its enzymatic center (Gloor et al, 2010). In contrast, the PCNA-FEN1 complex showed different conformations but appeared to be in the same functional state. Our findings suggest that FEN1 does not immediately release the 5′-flap after cleaving DNA, and the 3′ terminal nucleotide (at the +1 position) remains complementary to the template strand. This implies that product release and FEN1's departure from the DNA may be rate-limiting in lagging strand maturation. This notion is consistent with the kinetic data from the in vitro maturation process of human Okazaki fragments (Raducanu et al, 2022), and is also supported by our observations from sample purification and cryo-EM image processing that FEN1-containing complexes are dominant populations in the endogenous PCNA-related complexes. The association of RNaseH2 with PCNA induces movements to FEN1 and DNA in the PCNA-FEN1-RNaseH2 complex. In analogy, the entry of the next factor, LIG1, might facilitate the product release from FEN1 and the departure of FEN1 from the nicked site. Consistently, the recent in vitro FRET study indeed found that LIG1 efficiently releases the nicked product from FEN1 (Raducanu et al, 2022), and in the structure of the in vitro assembled PCNA-LIG1-FEN1 complex, FEN1 tilts away from the axis of PCNA and loses its DNA contact (Blair et al, 2022). Altogether, these data clearly show that these factors involved in the lagging strand processing are not merely competitors for the binding to PCNA and to the same substrate DNA. They likely undergo dynamic association-dissociation with the DNA, constituting an intricate network of DNA modulation events to drive the reactions towards the productive direction.

It is generally believed that PCNA acts as a platform for coordinating multiple enzymatic events of DNA replication, with the capacity to anchor up to three factors simultaneously. The structures of the endogenous PCNA-FEN1-RNaseH2 complex support this concept, confirming that this indeed occurs in vivo.

Together with the structural data of the in vitro assembled PCNA-FEN1-Pol δ and PCNA-FEN1-LIG1 complexes (Blair et al, 2022; Lancey et al, 2020), our data suggest an updated model for the molecular role of RNaseH2 in the Okazaki fragment maturation (Fig. 6A–L). The first role of RNaseH2 is dependent on its RNase activity. In this setting, RNaseH2 is recruited to PCNA, and moves together with the PCNA-Pol δ complex during the elongation of the Okazaki fragment. Once the PCNA-Pol δ reaches the RNA primer of the previous Okazaki fragment, RNaseH2 outcompetes Pol δ to recognize the hybrid RNA/DNA duplex and subsequently cleaves RNA residues in the primer, leaving the last RNA nucleotide in the 5′-end. This role of RNaseH2 is supported by the in vitro data that the affinity of human RNaseH2 for the RNA/DNA duplex containing 20 ribonucleotides is higher than 0.5 nM (Coffin et al, 2011). In sharp contrast, the DNA binding affinity of mammalian Pol δ is 70-fold lower (Einolf and Guengerich, 2000; Hedglin et al, 2016) (e.g., Kd of the human Pol δ for DNA is ~35 nM in the presence of PCNA) (Meng et al, 2010). After RNA digestion, RNaseH2 dissociates from the DNA but remains bound to PCNA through its PIP box. The second role of RNaseH2 is under-recognized, and is independent of its catalytic activity. During nick translation cycle, PCNA-bound Pol δ performs strand displacement to generate a short 5′-flap. For the same low-affinity reason, Pol δ could retract from the DNA and give way to RNaseH2, which repositions to the central axis of PCNA and binds to the upstream dsDNA, preventing the formation of the undesired 3′-flap. FEN1 is recruited through its PIP box to search for the 5′-flap. At this stage, RNaseH2, functioning as a dsDNA-binding protein, contributes to the FEN1-mediated recognition and removal of the 5′-flap. This role of RNaseH2 could be important in every round of nick translation, and the working unit of the primer removal might actually be a functional triad (PCNA-FEN1-RNaseH2-Pol δ). Interestingly, if we superimpose the PCNA-FEN1-Pol δ (Lancey et al, 2020) and our PCNA-FEN1-RNaseH2 complexes together using the PCNA monomer bound with FEN1 as the reference, the PIP boxes of Pol δ and RNaseH2 exactly occupy different PCNA monomers (Fig. EV5). Thus, these three factors should remain associated with PCNA during the entire nick translation process. After the cleavage of the last 5′-flap by FEN1, the recruitment of LIG1 through its PIP box onto PCNA promotes the product release and the departure of FEN1 from DNA (Blair et al, 2022). LIG1 completes the final task to fill the nick.

The structures of endogenous PCNA-FEN1 show that FEN1 binds above the PCNA ring, although FEN1 is biased to one side of the central axis of PCNA (Fig. 3). In contrast, in the structure of the in vitro reconstituted PCNA-FEN1-Pol δ complex, Pol δ is situated above the PCNA ring and FEN1 is highly flexible and free of DNA contact (Lancey et al, 2020). In fact, it would be impossible for both Pol δ and FEN1 to maintain stable binding to the DNA simultaneously. The reason is that during DNA synthesis, the 3′-OH end of the daughter strand is buried in the active center of Pol δ (Lancey et al, 2020), not compatible with the action of FEN1, which requires the exposure of 1-nt 3′-flap to be recognized by the hydrophobic wedge of FEN1 (Fig. EV5). Moreover, Pol δ cannot move backward to the upstream double-stranded DNA region without exercising its 3′–5′ exonuclease activity (Garg et al, 2004; Lancey et al, 2020). Therefore, during nick translation, for FEN1 to act, Pol δ has to dissociate from the DNA completely. Different from Pol δ, RNaseH2 could bind to the same DNA substrate with

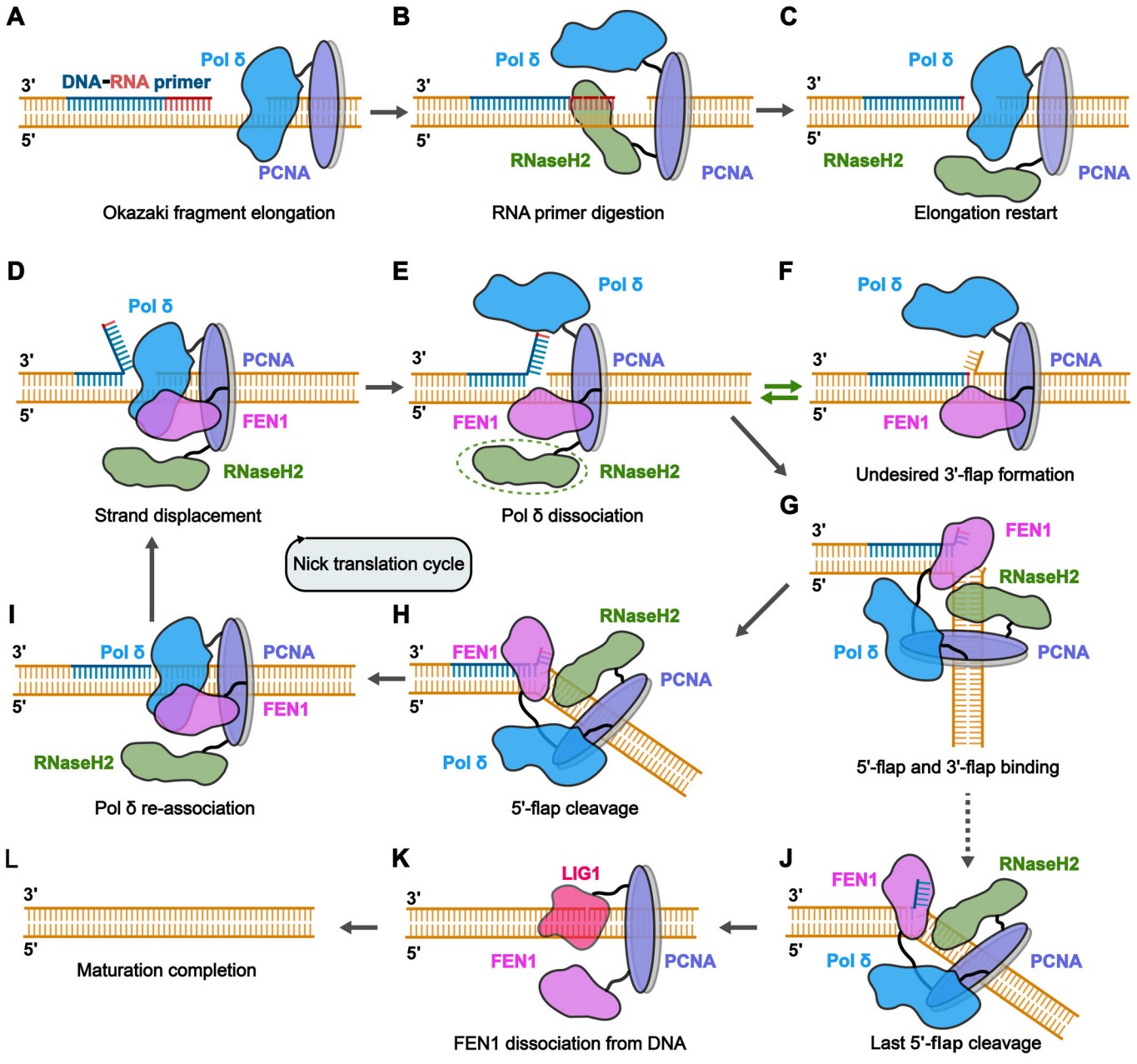

**Figure 6. Proposed molecular roles of RNaseH2 in the Okazaki fragment maturation.**

(A–C) The RNase activity-dependent role of RNaseH2 in the RNA primer removal. During the lagging strand synthesis, the PCNA-Pol δ complex elongates the Okazaki fragment in 5′–3′ direction, until it reaches the primer region of the previous Okazaki fragment (A). Pol δ has an intrinsically low affinity and frequently dissociates from the DNA. RNaseH2, recruited by PCNA, binds to the hybrid RNA/DNA duplex with a much higher affinity to digest RNA residues in the primer (B). RNaseH2 leaves one ribonucleotide at the 5′-end of the primer and is released from the DNA. Pol δ re-associates to the DNA to resume the elongation of the Okazaki fragment (C). (D–I) The catalytic activity-independent role of RNaseH2 in the nick translation cycle. Pol δ, FEN1, and RNaseH2 remain associated with PCNA during the nick translation. Pol δ invades the primer region to generate a short 5′-flap by strand displacement (D). Pol δ retract from the DNA, due to its low binding affinity (E). In the absence of RNaseH2, the 5′-flap might invade the upstream dsDNA to generate an undesired 3′-flap (F). In contrast, PCNA-bound RNaseH2 could bind to the upstream dsDNA with high affinity to prevent the formation of the 3′-flap. FEN1 searches for the 5′-flap, and by threading the 5′-flap, it recognizes and binds stably to both the 5′-flap and 1-nt 3′-flap (G). The 5′-flap is cleaved by FEN1 (H). In case of multiple rounds of nick translation, after FEN1 releases the cleaved product and dissociates from the DNA, Pol δ binds to the nick site and continues strand displacement (I). This nick translation process could repeat for several cycles until the primer is completely removed. (J) During the last round of nick translation, RNaseH2 facilitates the removal of the last 5′-flap by FEN1. (K) LIG1 is recruited to PCNA and displaces FEN1 for the DNA. (L) LIG1 completes the maturation of the Okazaki fragment.

FEN1 to form a stable complex on PCNA, and it does not compete for the binding of the 3′-flap on the daughter strand (Fig. EV5). Thus, the observed role of RNaseH2 as a dsDNA binding protein in the upstream DNA could only be fulfilled by RNaseH2, not by other factors. This function of RNaseH2 is not limited to the primer removal process of the Okazaki fragment maturation, and may be important for the ribonucleotide excision repair process as well, which also requires PCNA, Pol δ, RNaseH2 and FEN1 (Sparks et al, 2012). After the erroneous ribonucleotides embedded in the dsDNA is digested by RNaseH2, Pol δ fills the gap and performs strand displacement to generate a 5′-flap containing 1 nt RNA. The final cleavage of this 5′-flap by FEN1 may be as well promoted by RNaseH2 in the same mechanism. The dysfunction of RNaseH2 is closely linked to Aicardi–Goutières syndrome (AGS), which is a rare familial genetic disorder manifesting severe neurological impairment and multiple defects, mostly resulting in death in early childhood (Aicardi and Goutieres, 1984). Mutations in all three subunits of RNaseH2 have been found in patients with AGS (Crow et al, 2015; Crow et al, 2006). Of these, only 9% of AGS patients have RNaseH2A mutations, 68% have RNaseH2B mutations, and 23% have RNaseH2C mutations (Crow et al, 2015). It would be interesting to examine the impact of these mutation sites on the dsDNA binding ability and to verify whether the activity of FEN1 is affected.

Our structural data revealed a second PIP box in RNaseH2 on RNaseH2A subunit. This is reminiscent of LIG1, which also has two PIP boxes. One is located at the N-terminus and plays an important role in the initial recruitment to PCNA. The other PIP box is located in the DBD domain (DNA binding domain). In the structure of the in vitro assembled PCNA-LIG1 complex, the DBD PIP box binds to PCNA and this is the functional state where LIG1 is exerting ligase activity to ligate nicked DNA (Blair et al, 2022). For another example, Pol η has three PIP motifs, and they mediate different functions, such as the stimulation of DNA synthesis or the promotion of PCNA ubiquitination (Masuda et al, 2015). It is highly likely that the PIP motifs on two subunits of RNaseH2 may bind to PCNA in different stages and play different functions. RNaseH2B is located on the edge of the RNaseH2 trimer, away from DNA and the PIP box of RNaseH2B is exactly at its C-terminus, which is relatively more flexible (Bubeck et al, 2011). As a sequence-independent endonuclease, RNaseH2 cleaves the RNA-DNA hybrid at multiple sites (Chon et al, 2013; Figiel et al, 2011). When the PIP box of RNaseH2B is used as anchor to PCNA, RNaseH2 would have less spatial restraint to permit RNaseH2A to perform multiple enzymatic cleaving reactions. In contrast, in the PCNA-FEN1-RNaseH2 ternary complex, the attachment of the PIP box of RNaseH2A, which is closer to the central axis of PCNA, would greatly limits the flexibility of RNaseH2 to ensure that it now functions as a dsDNA binding protein. This could provide another level of regulation on the catalytic subunit RNaseH2A through controlling its access to different regions of substrate DNA. Further functional experiments are needed to verify the specific functions of these two PIP boxes.

In our samples of purified endogenous PCNA-containing complexes, we observed high abundance of DNMT1, CAF1, and histones. Eukaryotic Okazaki fragments are known to be similar in length of the DNA that makes up nucleosomes (Smith and Whitehouse, 2012). CAF1, a histone chaperone, assists in assembling new nucleosomes on daughter strands during DNA replication (Verreault et al, 1996). Therefore, it is likely that we have captured protein complexes involved in the lagging strand maturation coupled nucleosome assembly. Through 3D classification, we have resolved a few nucleosome-containing classes. Non-nucleosomal densities, possibly representing CAF1, DNMT1 or other factors, were also present in these classes. However, these particles showed high heterogeneity, making it challenging to identify associated factors through structural determination. Future investigations could benefit from a second affinity tag on factors of different processes to enrich protein complexes in specific pathways.

# Methods

**Reagents and tools table**

| Reagent/Resource | Reference or Source | Identifier or Catalog Number |
|---|---|---|
| **Experimental Models** | | |
| HEK-293 cells (*H. sapiens*) | ThermoFisher | A14528 |
| U-2 OS cells (*H. sapiens*) | ATCC | HTB-96 |
| **Recombinant DNA** | | |
| pMlink | Yigong Lab | N/A |
| pQlink | Addgene | 13667 |
| pET28a (+) | Addgene | 69864-3 |
| **Antibodies** | | |
| Mouse anti-PCNA | Proteintech | 10D10E11 |
| Rabbit anti-histone H4 | Beyotime | AF2581 |
| Goat anti-mouse Alexa Fluor™ 647 | Invitrogen | A-21235 |
| **Oligonucleotides and other sequence-based reagents** | | |
| Gel shift DNA | This study | Appendix Table S2 |
| **Chemicals, Enzymes and other reagents** | | |
| Hydroxyurea (HU) | Sigma | 127-07-1 |
| MNase | NEB | M0247S |
| Hoechst | Sigma | 23491-45-4 |
| Linear polyethylenimine (PEI), 25 kDa | Polysciences | 23966(1) |
| Lipofectamine 3000 | Thermofisher | L3000015 |
| Fetal bovine serum (FBS) | ThermoFisher | 26010074 |
| **Software** | | |
| GraphPad Prism 9.0 | | https://www.graphpad.com |
| ImageJ | | https://imagej.net/ |
| Relion | | https://relion.readthedocs.io/en/release-4.0/ |
| CryoSPARC | | https://cryosparc.com/ |
| Chimera | | https://www.cgl.ucsf.edu/chimera/ |
| ChimeraX | | https://www.cgl.ucsf.edu/chimerax/ |
| Coot | | https://www2.mrc-lmb.cam.ac.uk/personal/pemsley/coot/ |
| Phenix | | https://phenix-online.org/ |
| **Other** | | |
| 300 kV Cryo-Electron Microscope, Titan Krios | ThermoFisher | |

## Endogenous complex acquisition

The human PCNA gene was cloned into the PMlink vector with a twin-Strep-tag at the N-terminal position using *EcoR*I and *Xho*I restriction nuclease sites. Plasmids were isolated using a Magen Hipure Plasmid kit and then introduced into suspended HEK293 cells via PEI transfection. Initially, a small number of HEK293 cells were transfected for western blot analysis to confirm the expression of the PCNA protein. Subsequently, the HEK293 cell culture was scaled up, and the plasmids were transiently transfected into the cells when the cell density reached 1.8 million cells per ml. To induce cell cycle arrest at the S phase, 80 µM of hydroxyurea (HU) was added to the cells 24 h after plasmid transfection. Following this, 3 mM HU was introduced 16 h later to induce replication stress. Cells were harvested 6 h after the addition of the high concentration of HU. They were washed twice with PBS, rapidly frozen using liquid nitrogen, and stored at −80 °C for subsequent experiments.

To preserve the integrity of chromatin complexes during isolation, a gentle breaking method was employed. The protocol proceeded as follows. (1) Cell resuspension and lysis: Cells (1 L) were gently resuspended in Buffer A (10 mM HEPES pH 8.0, 10 mM KCl, 1.5 mM MgCl$_2$, 340 mM sucrose, 10% Glycerol, 1 mM DTT). Protease and phosphatase inhibitor cocktails were added to the suspension. Cell lysis was performed with 0.25% Triton X-100 for 45 min with continuous stirring using a magnetic stirrer. (2) Chromatin purification: Chromatin was carefully harvested by centrifugation at 5000 × g for 10 min at 4 °C. To remove cytoplasmic contaminants, the obtained chromatins were subjected to three rounds of resuspension and centrifugation using Buffer A. The purified chromatin was then resuspended in 20 mL of Buffer B (15 mM NaCl, 60 mM KCl, 10 mM HEPES pH 7.5, 2 mM CaCl$_2$). (3) MNase Digestion: MNase enzyme (5 kU/mL) was introduced to the chromatin to facilitate digestion and separation of chromatin complexes. The digestion process occurred at specific temperatures: room temperature for 10 min, 37 °C for 15 min, room temperature for 10 min, and finally at 4 °C for 30 min. The reaction was terminated with 10 mM EGTA. (4) Complex isolation: Following high-speed centrifugation at 23,700 × g for 30 min, the supernatant containing enzyme-cleaved complexes was carefully collected. Buffer C (15 mM NaCl, 60 mM KCl, 10 mM HEPES pH 7.5, 20% glycerol) was added to the sample to adjust glycerol levels. The sample was incubated with a 50% slurry of anti-Strep affinity beads 4FF (Smart-Lifesciences) for 3 h at 4 °C. PCNA and associated complexes bound to the beads were isolated using a gravity column, thoroughly washed with the wash buffer (15 mM NaCl, 60 mM KCl, 10 mM HEPES pH 7.5, 10% glycerol, protein inhibitor cocktail) to remove impurities, and subsequently eluted with the wash buffer containing 20 mM desthiobiotin.

To further separate complexes based on their sizes and compositions, a 10% to 30% glycerol gradient centrifugation step was conducted. The gradient was subjected to centrifugation in a Beckman SW41Ti Rotor for 12 h at a speed of 207,570 × g at 4 °C. Following centrifugation, 500 µL of liquid samples were carefully extracted from top to bottom for analysis. Subsequently, these samples were subjected to SDS-PAGE to assess the protein composition in various fractions along the density gradient. The components of the PCNA-related complexes were then identified through mass spectrometry analysis.

## Electron microscopy

To examine the endogenous complexes in different centrifugation fractions, corresponding samples were subjected to negative staining using uranyl acetate. The negatively stained grids were examined using a JEOL JEM-1400Flash electron microscope operating at 120 kV.

For cryo-EM sample preparation, sample fractions exhibiting high levels of both FEN1 and RNaseH2, as well as those rich in nucleosomes, were separately or jointly collected. For sample preparation, 4 µL of each sample was deposited onto glow-discharged holey-carbon grids (R1.2/1.3, Au, 300 mesh, Quantifoil), followed by blotting and flash freezing in liquid ethane utilizing an FEI Vitrobot Mark IV under conditions of 4 °C and 100% humidity. Subsequently, the grids were screened using an FEI Talos Arctica microscope operating at 200 kV to identify grids of high quality. Images were captured using a Titan Krios (FEI) microscope operating at 300 kV equipped with a K3 summit detector (Gatan) and a Gatan imaging filter (GIF) with a 20 eV slit. Movies were recorded using EPU software (ThermoFisher) at a magnification of 105,000× in super-resolution counting mode, resulting in a final pixel size of 0.415 Å at the object scale (super-resolution) and with defocus values ranging from −1.5 to −2.5 µm. Each movie consisted of 32 frames with a total of 59.5 electrons assigned, and the total exposure time was 2.6 s.

## Cryo-EM data processing

For the PCNA-FEN1 complex, four datasets with a total of 17,300 movies were collected. Each dataset was processed separately and good particles were pooled for additional rounds of 3D classification at a later stage. The movie stacks were imported into Relion 4.0 for local motion correction with dose weighting using MotionCor2 (Zheng et al, 2017). The parameters of the contrast transfer function (CTF) were estimated by CTFFIND4 (Zivanov et al, 2018). Initially, the Laplacian-of-Gaussian algorithm was used to pick particles, followed by multiple rounds of 2D classification. The resulting 2D classes exhibit large compositional heterogeneity, varying in sizes and features. Particles were then divided into groups (based on the sizes) and subjected to further 2D and 3D classifications separately (Fig. 2B–F). 3D classes of the PCNA-FEN1 complex were selected and pooled for 3D classification and refinement. The particles from good 3D classes were used as the training set, and another round of particle-picking was performed using Topaz (Bepler et al, 2019). The resulting particles were subjected to 2D classification, and the good classes were used for 3D analysis (the particles in the four batches of data were 371 K, 250 K, 67 K, and 28 K, respectively) (Appendix Fig. S2A). 3D classification revealed eight classes of different conformational states. Mask-based 3D classification and refinement were also applied to improve local densities of interested regions. Final rounds of refinement were done. CTF refinement and particle polishing further improved map resolution. The resolutions of the final maps were evaluated using the FSC curve at the cutoff of 0.143 (Appendix Fig. S2B). The local resolution of the density maps was measured using ResMap (Kucukelbir et al, 2014). Chimera (Pettersen et al, 2004) was used to display the local resolution (Appendix Fig. S2C).

The processing of the PCNA-FEN1-RNaseH2 complex particles followed similar procedures, involving particle-repicking using

Topaz (Bepler et al, 2019). After several rounds of 2D and 3D classification, around 59 K particles were kept for further analysis (Appendix Fig. S5A). 3D classification reported four major conformational states. After CTF refinement and Bayesian polishing, these classes were refined to final resolutions of 5.2–6.6 Å. To improve the local resolution of RNaseH2, the particles of state A, state B, state C, and state D were pooled and mask-based classification and refinement were performed, resulting in a local map for RNaseH2 at a resolution of 4.26 Å (Appendix Fig. S5A). Similarly, ResMap (Kucukelbir et al, 2014) was also used to evaluate the local resolution of the final density maps (Appendix Fig. S5C).

## Model modeling

For the PCNA-FEN1 complex, the crystal structures of PCNA (PDB ID: 1VYM) (Kontopidis et al, 2005) and the FEN1-DNA complex (PDB ID: 5UM9) (Tsutakawa et al, 2017) were used as initial references. Model adjustment and rebuilding were carried out manually using Coot (Afonine et al, 2018; Chen et al, 2010). The PIP box of FEN1 was built de novo, and DNA was modeled as A:T base pairs. For the PCNA-FEN1-RNaseH2 complex, the trimeric model of RNaseH2 predicted by AlphaFold2 (Jumper et al, 2021; Varadi et al, 2022) was used as the template. The C-terminal long loop, which harbors the PIP box of RNaseH2A was built de novo. The models were refined against the corresponding density maps using Phenix.real_space_refinement (Liebschner et al, 2019), with geometry restraints and secondary structure restraints applied. Model quality assessment was performed using MolProbity (Williams et al, 2018) (Appendix Table S1). Figure preparation was conducted using Chimera (Pettersen et al, 2004), ChimeraX (Meng et al, 2023), and PyMOL (http://pymol.org) to visualize and present the refined models of the complexes.

## Structural variability analysis

For Relion-based multibody refinement, particles of various states of the PCNA-FEN1 complex and the PCNA-FEN1-RNaseH2 complex were separately merged to create the initial 3D references. For the PCNA-FEN1 complex, PCNA and FEN1-DNA were treated as two rigid bodies, whereas in the PCNA-FEN1-RNaseH2 complex, PCNA, FEN1-DNA, and RNaseH2 were considered as three rigid bodies. Subsequently, masks were applied to perform multibody refinement using Relion (Nakane et al, 2018). The changes along the eigenvectors were visualized using Chimera (Pettersen et al, 2004) volume series movement (Movies EV2 and EV5). Different from multibody refinement, 3D variability analysis (Punjani and Fleet, 2021) utilized particles as input. The refined structure resulting from the merged particles served as a reference, and the entire structure was then masked. Within the 3D space, the particles were reconstructed into diverse structures based on distinct variance directions.

## Protein expression and purification

Human *PCNA* was cloned into pet28-a vector with 6xHis tag in the N-terminus. Human *RNaseH2A, RNaseH2B,* and *RNaseH2C* were constructed in the same pQlink vector with 6xHis tag in the N-terminus, respectively. The enzyme-inactive mutant *RNaseH2A* D34A/D169A (IM_DD/AA), *RNaseH2A* Y271A/F271A (PIPA_YF/AA), *RNaseH2B* F300A/F301A (PIPB_FF/AA), and the two PIP

boxes double mutant plasmid were generated in the wild type plasmid by PCR. The plasmids were transformed into *E. coli* strain BL21(DE3) cells. The bacteria were cultured at 37 °C until the $OD_{600}$ reached 0.8, 0.5 mM IPTG was added, and then the cells were cultured at 18 °C overnight. Cells were harvested by centrifugation at $3000 \times g$ for 10 min and suspended in lysis buffer containing 25 mM Tris pH 7.5, 150 mM NaCl, 10% Glycerol, 20 mM imidazole, 0.1% Triton X-100, 5 mM $MgCl_2$, EDTA free protease inhibitor and Benzonase (4 μg/mL). After cells were lysed by sonication, cell debris was removed by centrifugation at $20,000 \times g$ for 30 min. The supernatant was incubated with Ni-NTA resin for 10 min. After several rounds of washing using the lysis buffer, the bound proteins were eluted with the elution buffer containing 25 mM Tris pH 7.5, 150 mM NaCl, 10% Glycerol, 300 mM imidazole. The proteins were concentrated, diluted with low-salt buffer containing 25 mM Tris pH 7.5, 80 mM NaCl, 10% Glycerol and then loaded onto an anion exchanger, ManoQ column pre-equilibrated with the low-salt buffer. After washing with 10 column volumes, the proteins were eluted with high-salt buffer containing 25 mM Tris pH 7.5, 400 mM NaCl, 10% Glycerol. The protein sample was concentrated and loaded onto an Superdex 75 column pre-equilibrated with buffer containing 25 mM Tris pH 7.5, 150 mM NaCl for further purification. Fractions containing proteins were collected and flash-frozen in liquid nitrogen and stored at −80 °C in small aliquots.

## Gel electrophoresis mobility shift assay

Gel electrophoresis mobility shift assay was conducted in a reaction buffer containing 25 mM Tris pH 7.5, 100 mM NaCl, 10% Glycerol, 2 mM $MgCl_2$. When testing different lengths of dsDNA, 2.5 μM unlabeled DNA and different concentration of RNaseH2 (0.5, 1.5, 2.5, 5 μM) were incubated at RT for 30 min in a total of 20 μL. When comparing RNaseH2 binding to 20 bp dsDNA in the presence and absence of PCNA, the DNA was modified with 5′-FAM fluorophore. The reaction system was changed to 100 nM DNA, different concentrations of RNaseH2 (0.4, 0.6, 0.8, 1.0 μM) and the presence or absence of 100 nM PCNA. The entire reaction volume was loaded onto 5% native TBE-PAGE gels. The gels were run for 1 h at 4 °C at 80 V in TBE buffer. Gels were stained selectively with sybr gold and visualized using ChemiDoc™ Imaging System (BIO-RAD).

## Immunofluorescence microscopy

*GFP-RNaseH2A, mCherry RNaseH2B,* and *RNaseH2C* were constructed in the same pMlink vector and each has its own promoter and terminator. The mutant *RNaseH2A* D34A/D169A(IM_DD/AA), *RNaseH2A* Y271A/F271A (PIPA_YF/AA), *RNaseH2B* F300A/F301A (PIPB_FF/AA), and the two PIP boxes double mutant plasmid were generated in the wild type plasmid by PCR. U2OS cells were cultured on glass coverslips in six-well plate and transfected with these plasmids, respectively, through lipofectamine 3000 transfection reagent. At 48 h post-transfection, cells were stained with Hochest for 15 min. Then cellls were fixed in 4% paraformaldehyde for 15 min at RT and permeabilized with 0.1% Triton X-100. After eluting with PBS three times, the cells were blocked with 1% BSA in PBS for 1 h. Then the cells were incubated with PCNA antibody (Proteintech), washed with 1% BSA in PBS, incubated with Alexa fluor 647 secondary antibody (Invitrogen) and

washed with 1% BSA in PBS for another three times. Image was obtained using a confocal microscope (Nikon AXR laser scanning microscope).

## Data availability

The materials and data that support this study are available from the corresponding author upon reasonable request. Cryo-EM maps of endogenous state A to state H PCNA-FEN1 and state A to state D PCNA-FEN1-RNaseH2 complexes have been deposited in the Electron Microscopy Data Bank (EMDB, https://www.ebi.ac.uk/emdb/) with accession codes EMD-39342, EMD-39344, EMD-39346, EMD-39347, EMD-39348, EMD-39350, EMD-39351, EMD-39352, EMD-39358, EMD-39357, EMD-39355, EMD-39354, respectively. Atomic coordinates of state A to state H PCNA-FEN1 and state D PCNA-FEN1-RNaseH2 complexes have been deposited in the Protein Data Bank (PDB, https://www.rcsb.org/) with accession codes 8YJH, 8YJL, 8YJQ, 8YJR, 8YJS, 8YJU, 8YJV, 8YJW, 8YJZ, respectively.

The source data of this paper are collected in the following database record: biostudies:S-SCDT-10_1038-S44318-024-00296-x.

## Peer review information

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

    determination in RELION-3. Elife 7:e42166

## Acknowledgements

We thank the Core Facilities of Peking University School of Life Sciences for
assistance with negative-staining electron microscopy; the National Centre for
Protein Sciences at Peking University for assistance with mass spectrometry;
the Cryo-EM Platform of Peking University for cryo-EM data collection; the
High-performance Computing Platform of Peking University for help with
computation. This work was supported by the National Natural Science
Foundation of China (32321163647 to NG, 31830048 to QL), the National Key
Research and Development Program of China (2019YFA0508900), and the
Beijing Outstanding Young Scientist Program (BJJWZYJH01201910001005 to
QL). This work was also partially supported by the Ministry of Science and
Technology of China and Changping Laboratory.

## Author contributions

**Yuhui Tian**: Conceptualization; Data curation; Formal analysis; Validation;
Investigation; Visualization; Methodology; Writing—original draft;
Writing—review and editing. **Ningning Li**: Formal analysis; Supervision;
Investigation; Methodology. **Qing Li**: Conceptualization; Supervision;
Methodology. **Ning Gao**: Conceptualization; Resources; Formal analysis;
Supervision; Funding acquisition; Writing—original draft; Project
administration; Writing—review and editing.

Source data underlying figure panels in this paper may have individual
authorship assigned. Where available, figure panel/source data authorship is
listed in the following database record: biostudies:S-SCDT-10_1038-S44318-
024-00296-x.

## Disclosure and competing interests statement

The authors declare no competing interests.

# Expanded View Figures

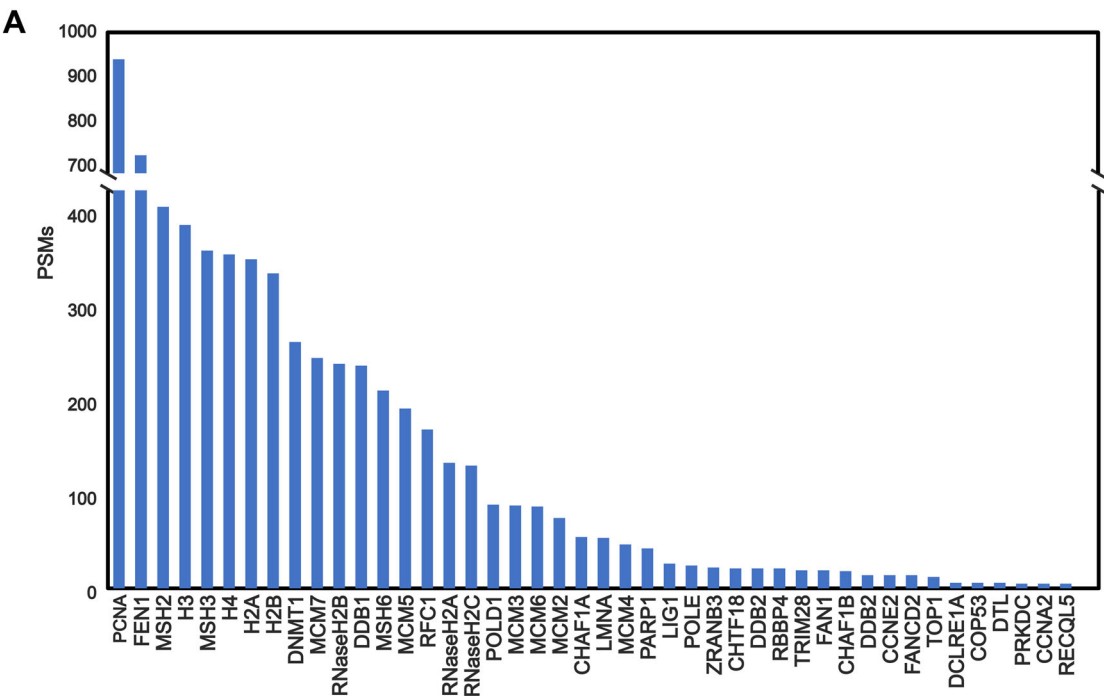

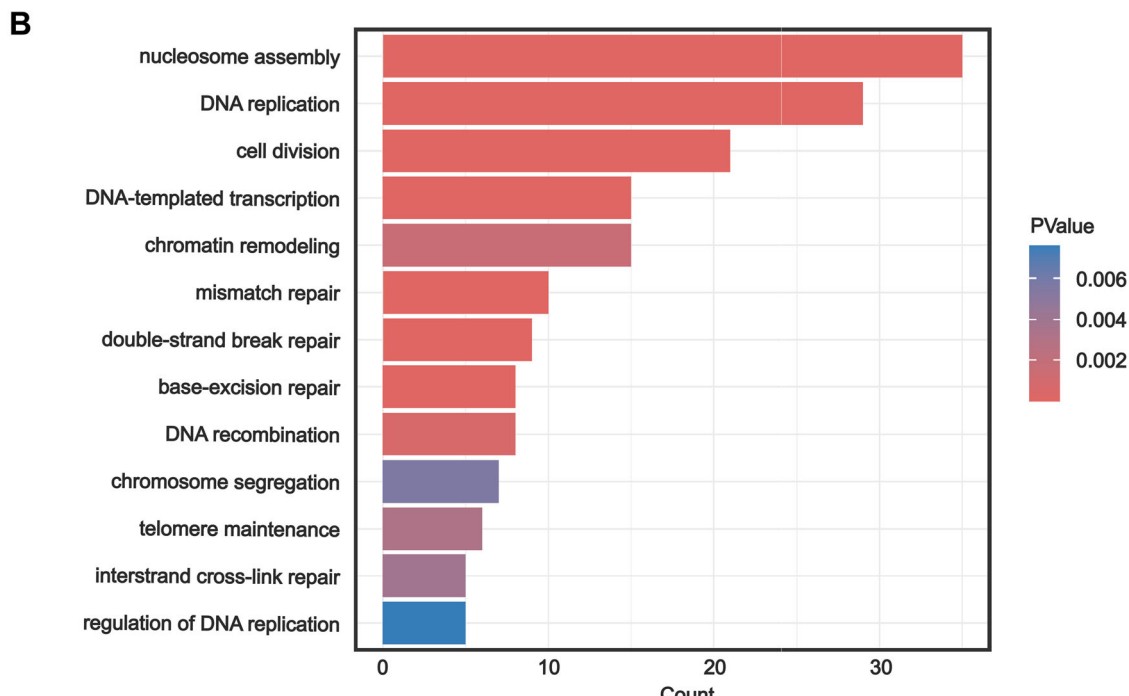

**Figure EV1.  Mass spectrometric analysis of the obtained endogenous PCNA-containing complexes.**

(A) A list of representative proteins related to PCNA identified in our samples by mass spectrometry. Candidate proteins were selected according to the BioGRID (Oughtred et al, 2021) database, and only proteins with the total number of identified peptide sequences (PSMs) greater than or equal to 5 were selected. (B) Gene ontology analysis of proteins identified by mass spectrometry according to the biological process. A total of 333 identified proteins were subjected to GO analysis. The *P*-values were calculated using hypergeometric distribution tests.

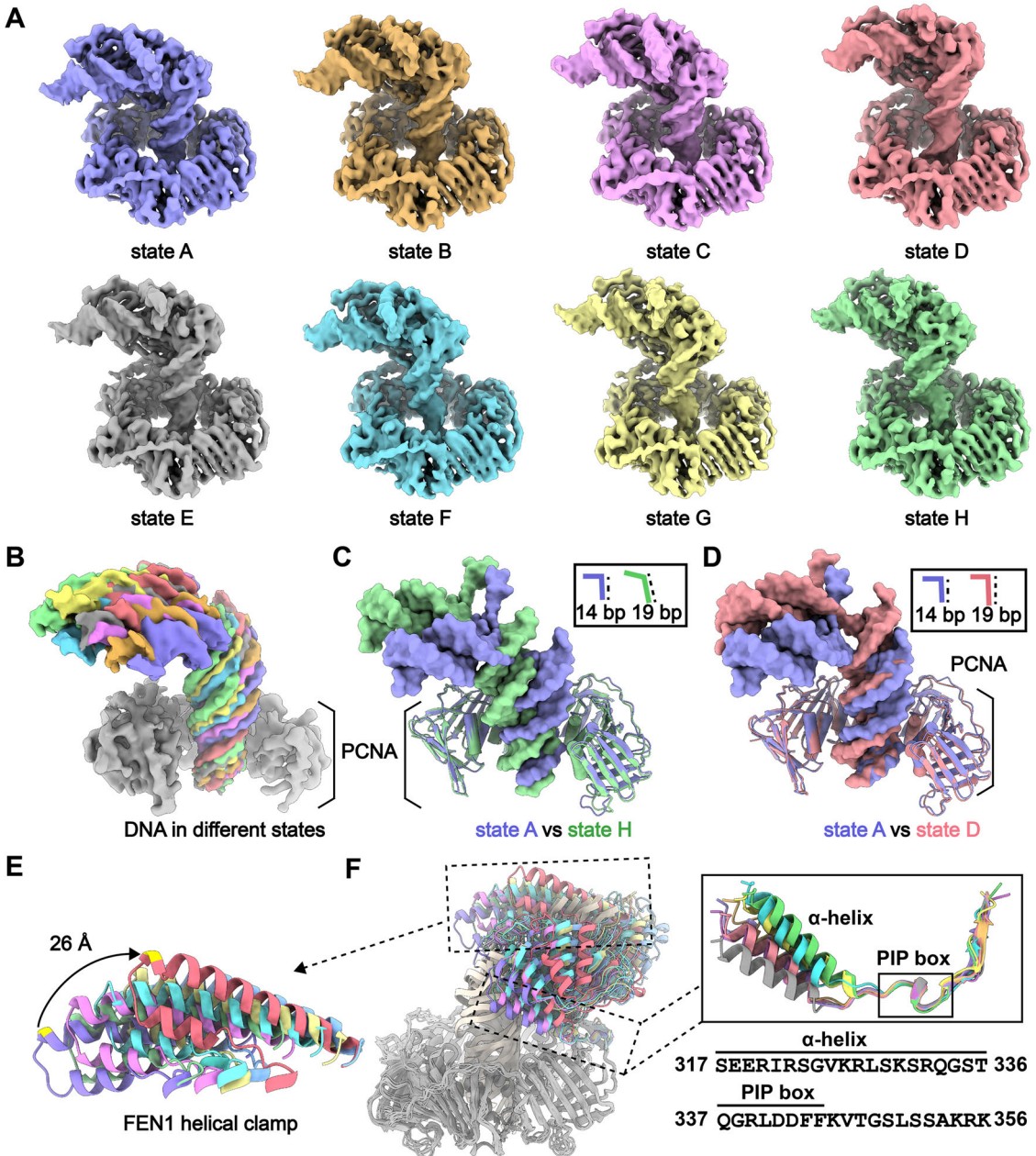

**Figure EV2. Different states of the PCNA-FEN1 complex.**

(**A**) Density maps of the eight conformational states. (**B**) Superimposition of the DNA in different states, with PCNA used as the reference of alignment. (**C**) Structural comparison of the DNA between state A and state H. The lengths of the upstream dsDNA in state A and state H are 14 bp and 19 bp, respectively. The bending angles of the DNA are significantly different between state A and state H. (**D**) Structural comparison of the DNA between state A and state D. The length of the upstream dsDNA in state D is 19 bp. The bending angles of the DNA in state A and state D are similar. (**E**) The positional change of the FEN1 helical clamp in different states, with PCNA used as reference of alignment. The distance between state A and state D is 26 Å measured by Cα of A116. (**F**) The C-terminus of FEN1 in different states. The conformation of the α-helix upstream of the PIP box has much greater changes than the PIP box.

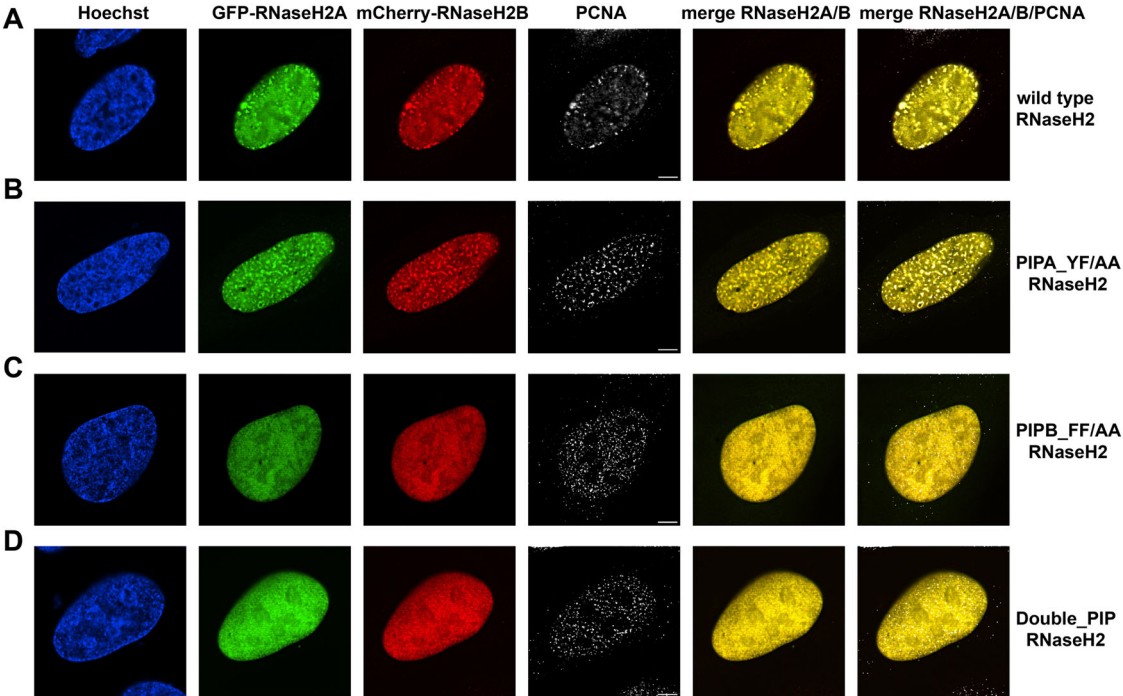

**Figure EV3. RNaseH2 depends on the PIP box of RNaseH2B to colocalize with PCNA.**

(A) GFP-RNaseH2A, mCherry-RNaseH2B colocalize with PCNA in U2OS cells. (B) Mutation of the PIP box in RNaseH2A does not affect its colocalization with PCNA. (C, D) GFP-RNaseH2A and mCherry-RNaseH2B distribute evenly in nuclei upon the mutation of RNaseH2B PIP box or double PIP box mutations in both RNaseH2A and RNaseH2B. PCNA was stained by immune-fluorescence with an Alexa fluor 647 secondary antibody. Scal bar, 5 μm.

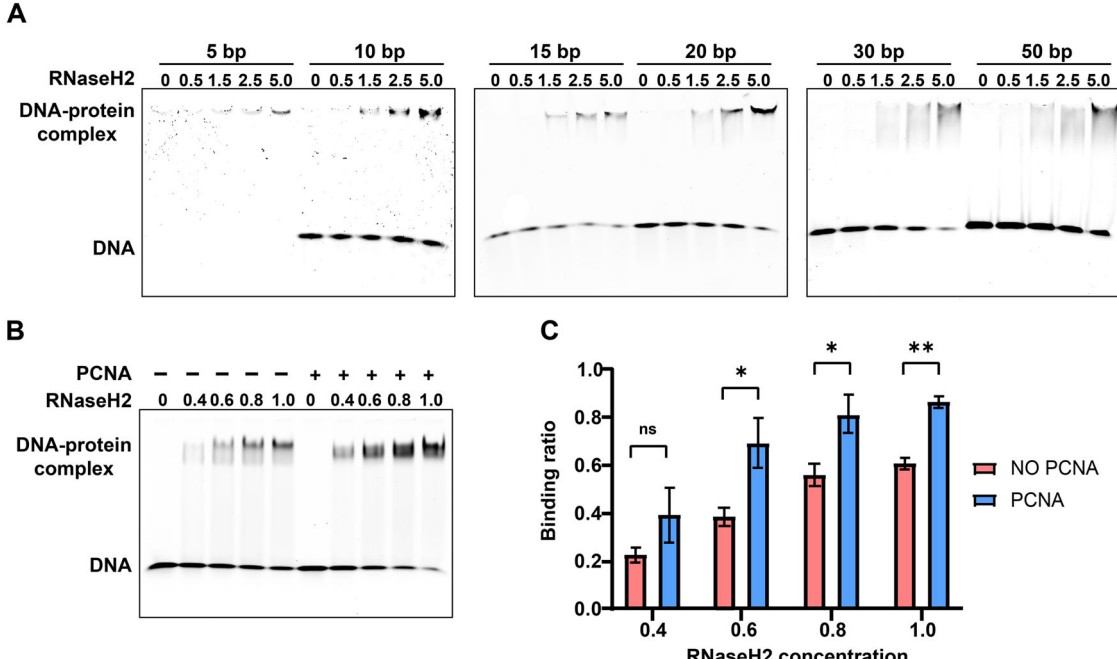

**Figure EV4. RNaseH2 binds strongly to the dsDNA of varying length.**

(A) RNaseH2 binds to the dsDNA in different length. The label-free dsDNA (2.5 μM) was incubated with RNaseH2 at different concentrations (0.5, 1.5, 2.5, 5 μM) for 30 min at RT and analyzed by 5% native TBE-PAGE gel. (B) PCNA promotes the binding of RNaseH2 to the dsDNA. 5′-FAM-labeled 20-bp dsDNA (100 nM) was incubated with RNaseH2 at different concentrations (0.4, 0.6, 0.8, 1.0 μM) in the presence or absence of PCNA (100 nM) for 30 min at RT and analyzed by 5% native TBE-PAGE gel. (C) Quantitative analysis of the binding of RNaseH2 to the dsDNA. The DNA-protein complexes in (B) were quantified using ImageJ, followed by statistical analysis using GraphPadPrism (n = 3 biological replicates). ns, not significant, *p < 0.05, **p < 0.001, multiple paired t tests. P values of PCNA vs NO PCNA: 0.092 (0.4 μM RNaseH2), 0.025 (0.6 μM RNaseH2), 0.010 (0.8 μM RNaseH2), 0.003 (1.0 μM RNaseH2). Error bars based on standard deviation (s.d.). Source data are available online for this figure.

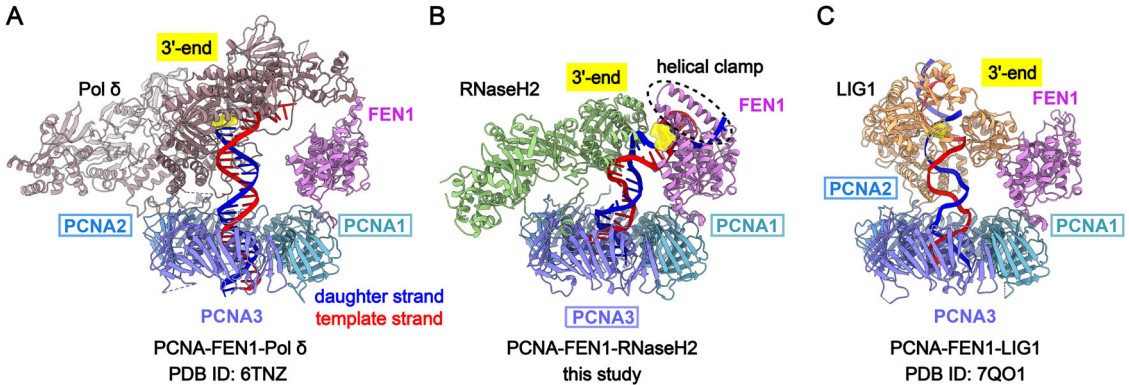

**Figure EV5.  Structural comparison of the PCNA-FEN1-Pol δ complex, the PCNA-FEN1-RNaseH2 complex, and the PCNA-FEN1-LIG1 complex.**

(A) The structure of the in vitro assembled PCNA-FEN1-Pol δ complex (PDB ID: 6TNZ) (Lancey et al, 2020). The PIP boxes of FEN1 and Pol δ docks onto the first and the second PCNA monomers (PCNA1 and PCNA2), respectively. The two strands of the DNA are separately colored. The 3′-end of the daughter strand is colored yellow and shown in surface representation. (B) The structure of the endogenous PCNA-FEN1-RNaseH2 complex. FEN1 and RNaseH2 bind to the first and the third PCNA monomers (PCNA1 and PCNA3), respectively. (C) The structure of the in vitro assembled PCNA-FEN1-LIG1 complex (PDB ID: 7QO1) (Blair et al, 2022). FEN1 and LIG1 bind to the first and the second PCNA monomers (PCNA1 and PCNA2), respectively. The three structures are displayed with PCNA monomer (PCNA1) as the reference of alignment.

