## [Peer Review File · The EMBO Journal]

Structural insight into Okazaki fragment maturation mediated by PCNA-bound FEN1 and RNaseH2

Yuhui Tian, Ningning Li, Qing Li, and Ning Gao

Corresponding author(s): Ning Gao (gaon@pku.edu.cn)

Review Timeline:

Submission Date:	24th Apr 24
Editorial Decision:	23rd May 24
Revision Received:	6th Oct 24
Editorial Decision:	22nd Oct 24
Revision Received:	27th Oct 24
Accepted:	28th Oct 24

Editor: Hartmut Vodermaier

Transaction Report:

Dr. Ning Gao
School of Life Sciences, Peking University
China

23rd May 2024

Re: EMBOJ-2024-117698-T

Structural profiling of endogenous PCNA-containing complexes provides insight into the Okazaki fragment maturation by FEN1 and RNaseH2

Dear Ning,

Thank you again for submitting your structural study of endogenous PCNA-bound complexes to The EMBO Journal. It has now been assessed by three expert referees, whose comments are copied below for your information. As you will see, the referees find the structural insights potentially interesting. However, referees 1 and 3 also feel that biochemical/functional follow-up investigations would be required to justify strong mechanistic conclusions and make the paper a good candidate for a broad general journal. Furthermore, all reviewers also raise a number of more specific technical/presentational/conceptual questions.

Should you be able to adequately address the referees' concerns, we would be interested in pursuing a revised version further for EMBO Journal publication. Since it is our policy to allow only a single round of major revision, it will however be important to carefully respond to all points at the time of resubmission. Should this require more time than our default three-months revision period, we would be happy to offer an extension, during which our 'scooping protection' (meaning that competing work appearing elsewhere in the meantime will not affect our considerations of your study) would of course remain valid.

Further information on preparing and uploading a revised manuscript can be found below and in our Guide to Authors. Thank you again for the opportunity to consider this work for The EMBO Journal, and I look forward to your revision.

With kind regards,

Hartmut

9) Digital image enhancement is acceptable practice, as long as it accurately represents the original data and conforms to community standards. If a figure has been subjected to significant electronic manipulation, this must be clearly noted in the figure legend and/or the 'Materials and Methods' section. The editors reserve the right to request original versions of figures and the original images that were used to assemble the figure. Finally, we generally encourage uploading of numerical as well as gel/blot image source data; for details see: embopress.org/page/journal/14602075/authorguide#sourcedata

At EMBO Press, we ask authors to provide source data for the main manuscript figures. Our source data coordinator will contact you to discuss which figure panels we would need source data for and will also provide you with helpful tips on how to upload and organize the files.

In the interest of ensuring the conceptual advance provided by the work, we recommend submitting a revision within 3 months (21st Aug 2024). Please discuss the revision progress ahead of this time with the editor if you require more time to complete the revisions. Use the link below to submit your revision:

Link Not Available

Referee #1:

Okazaki fragment processing is the final step in the replication of the DNA lagging strand. Tian et al report several cryoEM structures of protein complexes involved in this step, PCNA-FEN1 and PCNA-FEN1-RNaseH2. The postcatalytic structures of PCNA-FEN1 provide detailed insight into the interactions between FEN1 and PCNA, and FEN1 and the 5' and 3' flaps and are mostly similar to observations in prior FEN1 complex structures. The authors go on to describe the conformational flexibility in positioning the DNA and FEN1 with respect to the PCNA ring but the physiological significance of these various states remains unclear. The PCNA-FEN1-RNaseH2 structure is the most novel aspect of the manuscript and leads the authors to propose a catalytically independent role of RNaseH2 in nick translation, with RNaseH2 stabilizing the upstream DNA segment to prevent 3' flap formation. Although this model is potentially interesting, the resolution of these structures is quite low with 5-6 angstrom and limit the conclusions that can be confidently drawn from them. A major shortcoming of the work is that no biochemical evidence is provided to test any models and or conclusions derived from the static structures.

In summary, this is a potentially interesting paper for the DNA replication field if additional biochemical evidence is provided to support the non-canonical role of RNaseH2 during Okazaki fragment maturation.

Major comments:

1. The organization of the introduction could be improved. Paragraphs are very long and touch on many different aspects, which makes it hard for readers to follow.
2. The purification was done from cells transfected with a PCNA expression plasmid, presumably leading to substantial overexpression of PCNA. How do the authors ensure that the PCNA complexes they observe are not artificially driven by overexpression of PCNA? How do PCNA expression levels after transfection compare to physiological PCNA levels?
3. The authors only carry out a single affinity purification to enrich for PCNA-containing complexes. No negative control is provided to exclude background binding of proteins to affinity resin with untagged PCNA and to demonstrate specificity of the PCNA pulldown. This control is important if the authors want to draw any conclusions about PCNA complexes other than the ones they solved structures for.

4. The authors use HU to arrest cells, which stalls replication fork. It is unclear why this strategy was chosen and it is not explained in the main text. It is also unclear how HU may affect the structures.
5. The authors argue that RNaseH2 is a dominant band comigrating with PCNA on glycerol gradients starting from fraction 6 (including the fractions used for cryoEM) but no band is observed at the indicated position in the gel in fig 1b. It is also not explained what fractions were concentrated to get the 'con' sample.
6. In suppl fig 1, the authors do not include proteins labelled in fig 1b. Were some of them not detected by mass spec?
7. Based on low resolution EM classes, the authors argue that two PCNA rings can dimerize. It seems however that the particle density was very high on EM grids (fig. 2a), so the dimer classes could be artificially caused by proximity of two monomers on cryoEM grids. This would explain the low resolution obtained because the orientations of the monomers would vary. If the authors want to claim that PCNA can dimerize, this should be confirmed through additional experiments.
8. The authors state that they observe di- and trinucleosome arrangements as in the 30 nm fibre in their 2D class averages in suppl fig 2c. This is not supported by the data shown. Most class averages are rather featureless and the few classes that clearly show nucleosomes are single nucleosomes. Consequently, there is no basis for the claim that PCNA-related processes are involved in remodelling of chromatin structure.
9. The authors state that their PCNA-FEN1 structure is the first one in a physiological state. This is not true. Just because a structure is determined from assembling complexes with purified proteins doesn't mean the structure is not physiological. Indeed, many things the authors see in their structure have been observed previously in structures obtained with reconstituted FEN1-containing samples.
10. The cryoEM map region should be shown for residues in fig. 3i.
11. The authors observe several conformational states of the PCNA-FEN1 complex with slightly different orientations of the DNA and FEN1 with respect to PCNA. This flexibility is not surprising since molecules are not static but it remains unclear if there are any biological and functional consequences associated with these slightly different states.
12. Related to the previous point, since the relevance of the conformational states is unclear, the description of these states could be shortened.
13. The observation that a PIP box in RNaseH2A rather than RNaseH2B recruits RNaseH2 to Okazaki fragments and PCNA is potentially interesting but some form of biochemical or in vivo experiments should be provided to demonstrate that the interaction observed is actually involved in Okazaki fragment processing. This is important because the PIP box in RNaseH2B has been shown to localize RNaseH2 to replication foci. Is the same true for RNaseH2A's PIP box?
14. In fig 4g, it is unclear what PCNA-FEN1 state was used for the alignment with PCNA-FEN1-RNaseH2 complex. It is also unclear how correspondence between PCNA monomers in the trimer was established during alignments. It appears that the alignment is off by 60 degrees in the right panel.
15. The authors state that RNaseH2 is bound to double stranded DNA in their PCNA-FEN1-RNaseH2 structure. Given the low resolution, how can the authors be sure that RNaseH2 is not bound to a RNA-DNA hybrid? Although the authors argue that it must be DNA because the flaps have been formed, this statement is also not convincing based on the low resolution maps shown. Moreover, flaps could contain RNA if the primer has been partially displaced by pol delta.
15. On page 12, the authors argue that their structure of PCNA-FEN1-RNaseH2 overturns previous models that only show factors sequentially binding to PCNA during Okazaki fragment maturation. The authors neglect that there is precedent for multiple factors associating with PCNA at the same time, for example in the structure of PCNA-FEN1-Lig1 complex from the De Biasio group (Blair et al., Nat Communications 2022).
16. Binding of FEN1 and RNaseH2 simultaneously to PCNA does not necessarily mean there is a kinetic interdependence. If the authors want to claim this then some biochemical evidence should be provided.
17. In fig. 5, assigning the blue density as short flaps is not very convincing given the low resolution of the maps. It would make more sense to focus on the hydrophobic wedge and helical clamp as these are features that are easier to recognize at low resolution. If the authors want to claim that the flaps already have been generated in their PCNA-FEN1-RNaseH2 structure, some additional evidence should be provided. Related to that, showing DNA surface models in c is misleading. The EM map for DNA should be shown instead.
18. The model presented in fig 6 is confusing. In their structure, FEN1 and RNaseH2 bind the same side of the PCNA ring, but some model figures show them on opposite sides. The attachment of pol delta to the PCNA surface also changes in their models.

Minor comments:

19. The IDCL in fig 3d seems to be coloured light blue and not green.
20. Page 9: when describing 3Q8K in fig 3j, the nucleotide at the -1 position is unpaired and not "not complementary".
21. Suppl fig 5b shows a side view and not a top view of the PCNA-FEN1-DNA complex.
22. Why did the authors use the alphafold model rather than the crystal structure for RNaseH2 to build models?

Referee #2:

In the current report, Ning Gao, Qing Li and their colleagues have developed a strategy to purify endogenous PCNA-contained complexes from native chromatins and characterized the structures of two major species of the protein/DNA complexes (PCNA-FEN1 and PCNA-FEN1-RNase H2 complexes) using cryo-EM techniques. They have captured a series of snapshots of these two complexes, both of which are involved in the Okazaki fragment maturation processes of the lagging strand DNA synthesis.

Based on the abundance of the complexes, the authors have concluded that the product release from FEN1 is a rate-limiting step of Okazaki fragment maturation and confirmed the toolbelt function of PCNA as RNase H2 and FEN1 bind to different subunits of PCNA at the same time. They have also identified a previously unrecognized role of RNase H2, as a dsDNA binding protein, which promotes the 5'-flap cleaving activity of FEN1. All the revealed information is novel and represents a significant advance in the field. However, this reviewer suggests that the authors acknowledge and clarify the following technical limits and close the loose ends in the revised version of the manuscript:

1. The endogenous pull-down and Cryo-EM were only able to identify limited species of the protein DNA complexes. As the authors pointed out, in the process of Okazaki fragment maturation, there are at least four major enzymes that interact with PCNA including DNA polymerase delta, RNase H2, FEN1 and Lig 1. The major question in the field is how PCNA recruit these enzymes, in the fashion of toolbelt or being sequential in a timely manner. Because the other complexes were not identified maybe due to being very dynamic/transient, with the current data set, the authors wont be able to address such questions.
2. The pull-down products theoretically do not limit to the complexes involved in Okazaki fragment maturation process. In the lagging strand DNA synthesis, there are other important enzymes such as primase/DNA Polymerase alpha, which makes primers, DNA2 which also participates in primer removal. Their action and dynamic mechanism are what the field is waiting for. However, those complexes were not identified either through mass spectrometry or cryo-EM in the current study. Would synchronizing the cells in S-phase helps to enrich those complexes?
3. This reviewer was not clear if the authors intended to distinguish RNA from DNA to make the claim that RNase H2 binds to double strand DNA to promote FEN1 cleavage. Please clarify it.
4. In addition, the claim that RNase H2 moves from the 5' RNA end to 3' DNA end due to threading through the space between the DNA and DNA Polymerase delta is an imagination and based on the low affinity of DNA Pol delta. Additional evidence is required to support the claim.
5. Other two complexes including the nucleosome class and a two-ringed structure were also identified in the purification fractions 9-12. It was not clear to this reviewer if these two classes both contain PCNA and DNA and if they have anything to do with Okazaki fragment maturation. These two points need to be clarified.

Referee #3:

The paper by Tian et al reports a cryoEM study of PCNA complexes relevant to Okazaki fragment processing directly from mammalian cells, following transient transfection of HEK293 cells with Strep-tagged PCNA and recovery of PCNA complexes.

FINDING

The main finding of the paper is the recovery and identification of a PCNA-FEN1-RNaseH2-DNA complex, with both FEN1 and RNaseH2 bound via PIP boxes to the PCNA ring. The relative orientation of FEN1 (downstream) and RNaseH2 (upstream) on the DNA leads the authors to postulate a previously unrecognised role of RNaseH2, known to digest the RNA primer, in also assisting the 5'-flap cleaving activity of FEN1. This would be achieved by RNaseH2 binding and stabilisation of the dsDNA directly upstream of FEN1, thus preventing unwanted formation of a 3'-flap after nick translation by pol Delta.

ASSESSMENT

This study adds one more complex to the existing gallery of PCNA-based complexes that are known to be involved in OK fragment maturation, which includes binary and ternary PCNA complexes with Pol delta, FEN1 and Lig1, and the DNA substrate. The isolation and structure determination of the PCNA-FEN1-RNaseH2 complex on DNA is certainly a novel contribution to the field, and the mechanistic interpretation provided by the authors is interesting; however, it remains speculative without any of the follow-up functional experiments that could be thought of to support it (for instance, what is the affinity of RNaseH2 for dsDNA?).

An important issue that puzzles this reviewer is how come the authors could recover from cells and determine the high resolution structures of FEN1 and FEN1-RNaseH2 bound to PCNA, but failed to recover any of the other complex intermediates that are known to exist and have been characterised before, such as PCNA complexes with Pol Delta and/or FEN1 and/or Lig1? The authors should provide an explanation.

Referee #1:

Okazaki fragment processing is the final step in the replication of the DNA lagging strand. Tian et al report several cryoEM structures of protein complexes involved in this step, PCNA-FEN1 and PCNA-FEN1-RNaseH2. The postcatalytic structures of PCNA-FEN1 provide detailed insight into the interactions between FEN1 and PCNA, and FEN1 and the 5' and 3' flaps and are mostly similar to observations in prior FEN1 complex structures. The authors go on to describe the conformational flexibility in positioning the DNA and FEN1 with respect to the PCNA ring but the physiological significance of these various states remains unclear. The PCNA-FEN1-RNaseH2 structure is the most novel aspect of the manuscript and leads the authors to propose a catalytically independent role of RNaseH2 in nick translation, with RNaseH2 stabilizing the upstream DNA segment to prevent 3' flap formation. Although this model is potentially interesting, the resolution of these structures is quite low with 5-6 angstrom and limit the conclusions that can be confidently drawn from them. A major shortcoming of the work is that no biochemical evidence is provided to test any models and or conclusions derived from the static structures.

In summary, this is a potentially interesting paper for the DNA replication field if additional biochemical evidence is provided to support the non-canonical role of RNaseH2 during Okazaki fragment maturation.

Major comments:

1. The organization of the introduction could be improved. Paragraphs are very long and touch on many different aspects, which makes it hard for readers to follow.

We appreciate this suggestion. The introduction has been revised, with the following modifications: 1) The general introduction of DNA replication has been shortened; 2) the organization of the text has been improved to focus on PCNA and its role in lagging strand maturation.

2. The purification was done from cells transfected with a PCNA expression plasmid, presumably leading to substantial overexpression of PCNA. How do the authors ensure that the PCNA complexes they observe are not artificially driven by overexpression of PCNA? How do PCNA expression levels after transfection compare to physiological PCNA levels?

We fully understand this reviewer's concern. We believe that the overexpression had a neglectable effect to our structural study for a number of reasons. 1) We have controlled the overexpression condition by using only necessary amounts of plasmids for transfection. 2) During the sample purification, we have isolated chromatin fractions first and MNase digestion was used

to solubilize chromatin-associated protein complexes. PCNA proteins (or PCNA complexes) that are located in cytosol or not bound to chromatin were therefore removed and not included for structural analysis. 3) PCNA loading to chromatin requires specific loading proteins, RFC and RFC-like complexes (Majka & Burgers, 2004). Without overexpression of these loader proteins, excessive PCNA proteins could not be loaded onto DNA. 4) According to the gel image after glycerol density gradient centrifugation, the proportion of PCNA stably bound to chromatin does not appear to be significantly excessive compared with other proteins (from fraction 8 to heavier fractions, Figure 1b).

Therefore, we believe that for the two structurally resolved species of PCNA-containing complexes, the effect of overexpression is minimal. Both complexes contain associated DNA and should be snapshots of their physiological states.

3. The authors only carry out a single affinity purification to enrich for PCNA-containing complexes. No negative control is provided to exclude background binding of proteins to affinity resin with untagged PCNA and to demonstrate specificity of the PCNA pulldown. This control is important if the authors want to draw any conclusions about PCNA complexes other than the ones they solved structures for.

Our experimental design is to obtain chromatin-associated complexes containing PCNA for structural analysis. The mass spectrometry (MS) analysis of purified samples was a validation of their chromatin association, rather than an identification of new PCNA-interacting proteins.

In the revision, we have revised the text relevant to the results of the MS data to avoid drawing any premature conclusions.

4. The authors use HU to arrest cells, which stalls replication fork. It is unclear why this strategy was chosen and it is not explained in the main text. It's also unclear how HU may affect the structures.

One initial attempt of this study is to obtain large PCNA-containing complexes associated with DNA replication for structural analysis. Therefore, we have used HU treatment to arrest the cells in the DNA synthesis phase (S phase) such that the abundance of related complexes might be increased. Under our experimental condition, the cells in S phase have been increased from 50% to 80% upon HU treatment (Rebuttal Figure 1A-1C)

Rebuttal Figure 1. Sample purification from c.

(A, B) Flow cytometry analysis of cell cycle distribution using DAPI staining in the absence (A) or presence (B) of HU. (C) Quantitative analysis of cell cycle distribution. (D) Electrophoresis analysis of eluted samples prepared from cells with or without HU-treatment.

During sample preparation, we in fact have tried both conditions (with or without HU treatment). As shown in Rebuttal Figure 1D, SDS-PAGE analysis of the eluted samples after affinity purification showed no significant difference for the two conditions. Since the structural analysis was done with the samples from HU-treated cells, we have included HU-treatment in the Methods. HU-treatment is known to have an impact on the composition of the replisome. In this study, we report the structural analysis of two relatively small subcomplexes of the replisome, and the effect of HU on their structures should be minimal.

In the revision, we have clarified this in the Method.

5. The authors argue that RNaseH2 is a dominant band comigrating with PCNA on glycerol gradients starting from fraction 6 (including the fractions used for cryoEM) but no band is observed at the indicated position in the gel in fig 1b. It is also not explained what fractions were concentrated to get the 'con' sample.

In the manuscript, we stated that “For example, FEN1 co-migrates with PCNA in most of the fractions (starting from Fraction 4), and in the following heavier fractions (starting from Fraction

6), they were joined by DNMT1, MSH2, MSH3, MSH6, RNaseH2 subunits and four histone proteins.” We did not mean to claim that these factors all started to appear from Fraction 6 simultaneously. Their peaks were at different fractions. In Fraction 6, in addition to FEN1 and PCNA, high-molecular weight bands (between 140-200 kDa), as well as histone bands, started to appear. For RNaseH2 subunits, RNaseH2A and PCNA have similar molecular weight and are difficult to distinguish; RNaseH2C and histones H3/H2B also have similar molecular weight. RNaseH2B would be more apparent on the gel of silver staining (peaking at fractions 10-12).

Rebuttal Figure 2. Electrophoresis of protein samples after glycerol density gradient centrifugation.

The gel was stained using silver staining method. The peak of RNaseH2B is indicated by a red box.

In the revision, we have revised the text to avoid causing confusion to the readers.

The “con” sample was a concentrated sample from fractions 10-12, which was subjected to MS analysis.

6. In suppl fig 1, the authors do not include proteins labelled in fig 1b. Were some of them not detected by mass spec?

The proteins labelled in Figure 1b were all identified in the MS data. Because they were highly abundant in the sample, only for the purpose of display, they were not included in Supplementary Figure 1a. We have revised this figure panel (Rebuttal Figure 3).

Rebuttal Figure 3. A list of representative proteins related to PCNA identified in our samples by mass spectrometry. Candidate proteins were selected according to the BioGRID (Oughtred *et al*, 2021) database, and only proteins with the total number of identified peptide sequences (PSMs) greater than or equal to 5 were selected.

7. Based on low resolution EM classes, the authors argue that two PCNA rings can dimerize. It seems however that the particle density was very high on EM grids (fig. 2a), so the dimer classes could be artificially caused by proximity of two monomers on cryoEM grids. This would explain the low resolution obtained because the orientations of the monomers would vary. If the authors want to claim that PCNA can dimerize, this should be confirmed through additional experiments.

Figure 2a was prepared with one raw cryo-EM image of high defocus value to highlight the ring-like feature of PCNA particles. We agree that some of the dimer particles could be artificially caused by proximity of two monomers. But this could not explain all the dimer classes. In fact, after the three-dimensional classification, we found that there were at least three different forms of PCNA dimers (Rebuttal Figure 4). Although the densities of PCNA rings in Classes I and II are of low resolution, it is very clear that these two types of dimers are mediated by other proteins and/or DNA. In contrast, the dimer structure of Class III was determined at 8.7 Å resolution, which is sufficient for rigid-body docking to determine the polarity of the PCNA ring. Based on fitted high-resolution models, it appears that the IDCL motifs likely mediate the dimerization of the two PCNA rings. However, there is still a wobbling between the two rings; we were unable to further improve the resolution.

Rebuttal Figure 4. Structures of the dimeric PCNA complexes.

(A, B, C) Density maps of three dimer classes of PCNA-containing particles. (D) Atomic models of PCNA were fitted into the density map of the class III dimer.

8. The authors state that they observe di- and tri-nucleosome arrangements as in the 30 nm fiber in their 2D class averages in supp fig 2c. This is not supported by the data shown. Most class averages are rather featureless and the few classes that clearly show nucleosomes are single nucleosomes. Consequently, there is no basis for the claim that PCNA-related processes are involved in remodeling of chromatin structure.

The particle average images in supplementary figure 2c show the side view of the mono, di- or tri-nucleosomes. It can be seen that there is more than one nucleosome in some classes. The densities of the second and third nucleosome are unstable. The relative orientations of these nucleosomes show great similarities with native chromatin fiber in T lymphocytes cells (Hou *et al*, 2023) and *in vitro* recombinant chromatin 30-nm fiber (Soman *et al*, 2022; Song *et al*, 2014) (Rebuttal Figure 5).

We agree that it is highly speculative that PCNA-related processes are involved in chromatin remodeling. The suggestion is taken and we have revised the manuscript.

Rebuttal Figure 5. Comparison of the structural features of poly-nucleosomes.

(A) 2D classification images of di- or tri-nucleosome revealed in this study. (B) Tomographic slice of chromatin fibers in T-lymphoblasts cells. (C) Reconstituted 30-nm chromatin fiber containing 12 repeats. (D) Structure of the telomeric tetra-nucleosome. Panels B-D were from previous studies (Hou *et al.*, 2023; Soman *et al.*, 2022; Song *et al.*, 2014).

9. The authors state that their PCNA-FEN1 structure is the first one in a physiological state. This is not true. Just because a structure is determined from assembling complexes with purified proteins doesn't mean the structure is not physiological. Indeed, many things the authors see in their structure have been observed previously in structures obtained with reconstituted FEN1-containing samples.

We thank the reviewer for this suggestion. Our initial intention is to emphasize that this is the first structure of endogenously purified PCNA-FEN1 complex. We have revised the text according to the suggestion.

10. The cryoEM map region should be shown for residues in fig. 3i.

The cryoEM density map is added in figure 3i and the more density maps on the DNA are shown in expanded figure 4.

11. The authors observe several conformational states of the PCNA-FEN1 complex with slightly different orientations of the DNA and FEN1 with respect to PCNA. This flexibility is not surprising since molecules are not static but it remains unclear if there are any biological and functional consequences associated with these slightly different states.

We have measured these changes quantitatively. As shown in expanded figure 6, along with the change in DNA angle increased by 20°, the relative displacement of the 5'-flap and the 3'-flap

could be as large as 30 Å. Although we cannot clearly explain the functional significance of these changes. But these changes suggest that FEN1 binding to DNA *in vivo* does not require a specific length on the upstream dsDNA for FEN1 to exert its catalytic function.

12. Related to the previous point, since the relevance of the conformational states is unclear, the description of these states could be shortened.

The suggestion is well taken. We have shortened the discussion on the conformational plasticity of these two complexes.

13. The observation that a PIP box in RNaseH2A rather than RNaseH2B recruits RNaseH2 to Okazaki fragments and PCNA is potentially interesting but some form of biochemical or *in vivo* experiments should be provided to demonstrate that the interaction observed is actually involved in Okazaki fragment processing. This is important because the PIP box in RNaseH2B has been shown to localize RNaseH2 to replication foci. Is the same true for RNaseH2A's PIP box?

We thank the reviewer for this suggestion. To address this question, we have performed two sets of experiments.

(1) We purified wild-type and four types of mutant RNaseH2 complex, containing enzymatically dead mutant RNaseH2A subunit (IM_DD/AA), single PIP-box mutant of RNaseH2A or RNaseH2B subunit [PIPA (YF/AA) and PIPB (FF/AA)], and double PIP-box mutant (both RNaseH2A and RNaseH2B subunits), and incubated them with PCNA *in vitro* and performed size-exclusion chromatography using Superdex 200 (Rebuttal Figure 6). The IM_DD/AA mutant showed no defect in binding to PCNA, with the PCNA-RNaseH2 complex peaking at 14.94 ml elution volume. Single mutants of RNaseH2 (PIPA and PIPB) could still bind to PCNA, but the peaks were now at 15.14 ml, suggesting that their binding is less stable. In contrast, when both PIP boxes were mutated, RNaseH2 no longer interacts with PCNA.

(2) To explore the function of RNaseH2A PIP boxes, we constructed a plasmid encoding all three subunits GFP-RNaseHA, mCherry-RNaseH2B and RNaseH2C and transiently transformed the plasmid into U2OS cells. We found that RNaseH2A, RNaseH2B and PCNA co-localized at the replication site in wild-type and RNaseH2A PIP box mutant cells (Rebuttal Figure 7). However, when the RNaseH2B PIP box was mutated, RNaseH2A and RNaseH2B were almost evenly distributed in the nucleus and did not co-localize with PCNA. This result is consistent with previous work (Bubeck *et al*, 2011). It might suggest that the initial recruitment of RNaseH2 to the replication site does not rely on RNaseH2A PIP box but mainly relies on the interaction between RNaseH2B PIP box and PCNA.

Nevertheless, we acknowledge that the function of RNaseH2A PIP box remains to be explored. Given that the focus of this manuscript is on the structural analysis of the

PCNA-FEN1-RNaseH2 complex, further in-depth functional examination of RNaseH2A PIP box would be beyond the scope of the current manuscript and lead to significant delay to the publication of the structural data.

Rebuttal Figure 6. Both RNaseH2A PIP box and RNaseH2B PIP box contribute to the binding to PCNA.

(A) Purified proteins of PCNA, wild-type and mutant RNaseH2 complexes. (B-F) Size-exclusion chromatography of PCNA, RNaseH2 and their complexes. PCNA and RNaseH2 were incubated at a molar ratio of 1:1 for 30 min at RT and then loaded onto Superdex 200 for analysis. (G) Protein electrophoresis of selected fractions (12-25) in the panels B-F.

Rebuttal Figure 7. RNaseH2 depends on the PIP box of RNaseH2B to colocalize with PCNA.

(A) GFP-RNaseH2A, mCherry-RNaseH2B colocalize with PCNA in U2OS cells. (B) PIP box mutant of RNaseH2A does not abolish the recruitment of RNaseH2 to the PCNA foci. (C, D) GFP-RNaseH2A and mCherry-RNaseH2B distribute evenly in nuclei upon the mutation of RNaseH2B PIP box or double PIP box mutations in both RNaseH2A and RNaseH2B. PCNA was stained by immune-fluorescence with an Alexa fluor 647 secondary antibody. Scale bar, 5 μ m.

14. In fig 4g, it is unclear what PCNA-FEN1 state was used for the alignment with PCNA-FEN1-RNaseH2 complex. It is also unclear how correspondence between PCNA monomers in the trimer was established during alignments. It appears that the alignment is off by 60 degrees in the right panel.

For Fig. 4g, the state D of the PCNA-FEN1 complex was aligned with the PCNA-FEN1-RNaseH2 complex, using the PCNA monomer bound with FEN1 as the reference for alignment. The right and middle panels were derived from the same structural alignment. We have changed the coloring scheme on the PIP boxes in the right panel, to reflect the fact that the PIP boxes of FEN1 in the two different complexes can be well aligned.

15. The authors state that RNaseH2 is bound to double stranded DNA in their

PCNA-FEN1-RNaseH2 structure. Given the low resolution, how can the authors be sure that RNaseH2 is not bound to a RNA-DNA hybrid? Although the authors argue that it must be DNA because the flaps have been formed, this statement is also not convincing based on the low-resolution maps shown. Moreover, flaps could contain RNA if the primer has been partially displaced by pol delta.

Due to the low resolution, it is currently impossible to structurally confirm that the location is DNA. However, based on the current spatial topology, it can be confirmed that the location where RNaseH2 binds is a double-stranded DNA region.

According to the current structure, it can be determined that the template strand DNA is complete without a gap, while the newly synthesized strand is bent by FEN1, with a break in the middle, and the total length is greater than the template strand. The 5' end of the break forms a flap

of at least 2 nt, and the 3' end of the break forms a flap of 1 nt (Rebuttal Figure 8)

Rebuttal Figure 8. Diagram of the DNA topology in the PCNA-FEN1-RNaseH2 complex.

(A) The schematic diagram of the DNA in the PCNA-FEN1-RNaseH2 complex. The template strand is colored in blue and the nascent strand in black. (B) Direction of DNA polymerase along the nascent strand during the lagging strand synthesis. The two adjacent Okazaki fragments are shown.

These structural features indicate that Pol δ should have undergone strand displacement. Since Pol δ moves along the 5'-3' direction in the nascent strand, this also means that Pol δ has already added at least 3 deoxyribonucleotides to the 3'-end of the current Okazaki fragment. Therefore, the 3' end of the break and the upstream region should be all DNA. RNaseH2 binds to the double-stranded region upstream of the 3' end of the break.

Secondly, at the joint site of the two Okazaki fragments, RNA is only located at the RNA primer of the previous Okazaki fragment, which should be in the region downstream of the 5' end of the break. If Pol δ only performs partial displacement of the RNA primer, the remaining RNA should be in the 5'-end region of the break. But the binding position for RNaseH2 is located in a

different direction of the break. Therefore, there should be no RNA in the region where RNaseH2 binds.

16. On page 12, the authors argue that their structure of PCNA-FEN1-RNaseH2 overturns previous models that only show factors sequentially binding to PCNA during Okazaki fragment maturation. The authors neglect that there is precedent for multiple factors associating with PCNA at the same time, for example in the structure of PCNA-FEN1-Lig1 complex from the De Biasio group (Blair et al., Nat Communications 2022).

This appears to be a misunderstanding of our statement. In the manuscript, we stated that “In the previous models (for example, see ref.(Sun *et al*, 2023)), these factors were depicted to act one by one to catalyze the maturation of the lagging strand. Our structure of the PCNA-FEN1-RNaseH2 complex show that the same DNA substrate could stably bind to two enzymes, suggesting the presence of certain kinetic inter-dependence for these two factors.” In our manuscript, we would like to make a distinction between the binding to PCNA and the binding to the DNA substrate. The *in vitro* reconstituted complexes show that PCNA could bind to two factors at the same time. However, in the structures of these reconstituted complexes, including PCNA-Pol δ -FEN1 and PCNA-Lig1-FEN1, only one protein factor is bound with DNA. Although they remain bound to PCNA simultaneously, they still catalyze their DNA-related reactions one by one and independently. We emphasize that FEN1 and RNaseH2 not only bind to the same PCNA, but also to the same DNA substrate. This is also the reason why we speculate that the two factors may regulate each other through modulating the conformations of the DNA, which is not recognized in previous models.

17. Binding of FEN1 and RNaseH2 simultaneously to PCNA does not necessarily mean there is a kinetic interdependence. If the authors want to claim this then some biochemical evidence should be provided.

As we have stated in our response to the comment #15, based on the structural observation that both factors bind to the same DNA substrate, we propose a possible role of RNaseH2 in promoting the *in vivo* function of FEN1 in cleaving 5' flap DNA. We have thought about to perform additional experiments to test this hypothesis.

Firstly, it is very difficult to study the enzyme kinetics using a cell-based assay, because FEN1 is not only involved in the maturation of Okazaki fragments, but also in long-patch base excision repair, telomere maintenance, and stalled replication fork rescue (Balakrishnan & Bambara, 2013). The enzymatic activity of flap cleavage may be regulated by different proteins in

different regions of the chromatin. RNaseH2 is also a multifunctional protein complex. In addition to participating in the processing of Okazaki fragment primers, it also plays an important role in the removal of the R loop (Hyjek et al, 2019). Knocking out or knocking down RNaseH2 subunits in cells may affect the stability of the genome and change a variety of cell phenotypes (Bartsch et al, 2017; Ghosh et al, 2022). Therefore, there is no simple way to correlate these experimental outcomes with the kinetic characteristics of FEN1 in the lagging strand maturation. There are also technical limitations to monitoring intracellular FEN1 dynamics. It is relatively difficult to directly detect the 2-10 nt DNA of FEN1 products in the cell nucleus. The product fragments are too short and have great heterogeneity, making it difficult to detect with probes. In addition, this short ssDNA may be produced by other physiological processes, including by nucleases such as EXO1 (Keijzers et al, 2016) and CtIP (Sartori et al, 2007).

Secondly, given the difficulty in exploring FEN1 kinetics in cells, we have attempted to investigate the effect of RNaseH2 on FEN1 cleavage efficiency *in vitro*. We found that there are a few technical problems that are difficult to deal with. (1) We have purified FEN1, wild-type and enzyme inactive mutant RNaseH2 proteins and found that FEN1 has high enzyme activity efficiency and fast reaction rate (Rebuttal figure 9), which is consistent with previous reports [Kcat/Km is about $1.5 \times 10^8 \text{M}^{-1} \text{s}^{-1}$ ($9 \text{nM}^{-1} \text{min}^{-1}$) using double flap DNA substrate] (Liu et al, 2006; Tsutakawa et al, 2011). (2) RNaseH2 randomly binds dsDNA *in vitro*, and only 5-10 bp is sufficient for its binding (Rebuttal Figure 10). It is difficult to control where RNaseH2 binds on the dsDNA. If RNaseH2 could freely bind to all possible regions of the DNA substrate, one would expect an inhibitory effect on the 5' flap cleavage by FEN1 (Rebuttal figure 9), simply because of their direct competition on the dsDNA region. (3) We have also tried to add PCNA in the system to position RNaseH2 to the desired upstream region of the DNA substrate. PCNA can slide in or out from both ends of the DNA (Craggs et al, 2014; De March et al, 2017). We have tried to add neutravidin to bind to the biotin-modified DNA at one end to prevent PCNA from sliding in from there, but we could not control the polarity of the loaded PCNA on the other end. To make it worse, the loaded PCNA ring is also very dynamic and could slip away from the DNA.

Rebuttal Figure 9. RNaseH2 slightly inhibits the enzymatic activity of FEN1 *in vitro*, in absence

of the PCNA mediated positioning effect

(A, B) Urea denaturing gel electrophoresis of the product DNA under different reaction conditions. The DNA concentration is 100 nM. The protein concentration is indicated in the figure in nM. Wild-type RNaseH2 was added to the reaction system in Panel A, and the enzymatically inactive mutant RNaseH2 was added to the reaction system in Panel B. The reaction was terminated after 10 min at room temperature. The flap-containing DNA substrate was obtained by annealing three single-stranded DNA strands after denaturation at 95°C and purified by native-PAGE, which would presumably contain a mixture of 5' flap and 3' flap. 5' -FAM is labelled at the end of the 5' flap.

Rebuttal Figure 10. RNaseH2 binds strongly to the dsDNA of varying length.

(A) RNaseH2 binds to the dsDNA in different length. The label-free dsDNA (2.5 μM) was incubated with RNaseH2 at different concentrations (0.5, 1.5, 2.5, 5 μM) for 30 min at RT and analyzed by 5% native TBE-PAGE gel. (B) PCNA promotes the binding of RNaseH2 to the dsDNA. 5'-FAM-labeled 20-bp dsDNA (100 nM) was incubated with RNaseH2 at different concentrations (0.4, 0.6, 0.8, 1.0 μM) in the presence or absence of PCNA (100 nM) for 30 min at RT and analyzed by 5% native TBE-PAGE gel. (C) Quantitative analysis of the binding of RNaseH2 to the dsDNA. The DNA-protein complexes in panel b were quantified using ImageJ, followed by statistical analysis using GraphPadPrism. ns, not significant, * $p < 0.05$, ** $p < 0.001$, multiple paired t tests.

Given the tedious procedures in assembling PCNA, RNaseH2, FEN1 on the same DNA and the extremely low yield of the desired, “correct” complex for measuring the *in vitro* reaction, an ideal (possibly only) way is to perform single-molecule experiments with high temporal resolution.

However, single-molecule experiments require special equipment and expertise, which we do not have currently, and will take a long time to set up. These experiments could not be finished in the timeframe of this revision. This manuscript is focusing mainly on structural analysis of the endogenous complexes. We wish to carry out these experiments in a follow-up study.

18. In fig. 5, assigning the blue density as short flaps is not very convincing given the low resolution of the maps. It would make more sense to focus on the hydrophobic wedge and helical clamp as these are features that are easier to recognize at low resolution. If the authors want to claim that the flaps already have been generated in their PCNA-FEN1-RNaseH2 structure, some additional evidence should be provided. Related to that, showing DNA surface models in c is misleading. The EM map for DNA should be shown instead.

The density of flap DNA is visible in states C and D of the PCNA-FEN1-RNaseH2 complex. After hiding the density maps of the helical clamp and hydrophobic wedge, the density of DNA is clear, as shown in Rebuttal Figure 11A. The conformational changes of FEN1 and DNA are coordinated, so we marked the flap DNA in the figure 5, although its resolution is poor. In order to clearly show the difference between DNA in state A and state D, we used the surface map of the atomic model.

We have taken this suggestion and this panel is now prepared with actual density maps (Rebuttal Figure 11B).

Rebuttal Figure 11. Density maps of the PCNA-FEN1-RNaseH2 complex.

(A) Segmented density maps after hiding helical clamp of FEN1. (B) Comparison of the DNA configurations in the density maps of states A and D using PCNA as the reference for alignment.

19. The model presented in fig 6 is confusing. In their structure, FEN1 and RNaseH2 bind the same side of the PCNA ring, but some model figures show them on opposite sides. The attachment of pol delta to the PCNA surface also changes in their models.

We thank the reviewer for this suggestion. To avoid confusion, we have revised the model. FEN1, Pol delta and RNaseH2 all bind to the same side of PCNA (Rebuttal Figure 12).

Rebuttal Figure 12. Proposed molecular roles of RNaseH2 in the Okazaki fragment maturation.

Minor comments:

20. The IDCL in fig 3d seems to be colored light blue and not green.

The color of the IDCL in figure 3d appears to be light blue because it is under the transparent surface. The text has been modified.

21. Page 9: when describing 3Q8K in fig 3j, the nucleotide at the -1 position is unpaired and not "not complementary".

The text has been modified.

22. Supp fig 5b shows a side view and not a top view of the PCNA-FEN1-DNA complex.

The text has been modified.

23. Why did the authors use the alphafold model rather than the crystal structure for RNaseH2 to build models?

Because some sequences are missing in the crystal structure, the amino acid sequence of the structure predicted by alphafold is complete. We also compare the structure of alphafold prediction with the crystal structure, and find that there is little difference in the stable region of the structure. So, we use the structure predicted by alphafold for model building, which is more complete.

Referee #2:

In the current report, Ning Gao, Qing Li and their colleagues have developed a strategy to purify endogenous PCNA-contained complexes from native chromatin and characterized the structures of two major species of the protein/DNA complexes (PCNA-FEN1 and PCNA-FEN1-RNase H2 complexes) using cryo-EM techniques. They have captured a series of snapshots of these two complexes, both of which are involved in the Okazaki fragment maturation processes of the lagging strand DNA synthesis. Based on the abundance of the complexes, the authors have concluded that the product release from FEN1 is a rate-limiting step of Okazaki fragment maturation and confirmed the toolbelt function of PCNA as RNase H2 and FEN1 bind to different subunits of PCNA at the same time. They have also identified a previously unrecognized role of RNase H2, as a dsDNA binding protein, which promotes the 5'-flap cleaving activity of FEN1. All the revealed information is novel and represents a significant advance in the field. However, this reviewer suggests that the authors acknowledge and clarify the following technical limits and close the loose ends in the revised version of the manuscript:

1. The endogenous pull-down and Cryo-EM were only able to identify limited species of the protein DNA complexes. As the authors pointed out, in the process of Okazaki fragment maturation, there are at least four major enzymes that interact with PCNA including DNA polymerase delta, RNase H2, FEN1 and Lig 1. The major question in the field is how PCNA recruit these enzymes, in the fashion of toolbelt or being sequential in a timely manner. Because the other complexes were not identified maybe due to being very dynamic/transient, with the current data set, the authors won't be able to address such questions.

We agree with the reviewer that our study did not provide any information on Pol delta and Lig1. In fact, we detected Lig1 and Pol delta in mass spectrometry, but the abundances of these proteins were relatively low (compared with FEN1), and no obvious protein bands were observed in gel with coomassie brilliant blue staining. In the subsequent cryo-EM data analysis, only two stable complexes (PCNA-FEN1; PCNA-FEN1-RNaseH2) were resolved at resolutions of meaningful interpretation. There are two possible reasons: (1) FEN1-mediated reaction is rate-limiting, and therefore, PCNA-pull down has greatly enriched FEN1-containing complexes over Pol delta and Lig1; (2) as suggested by the reviewer, Pol delta or Lig1-containing complexes could be very dynamic, and 2D and 3D classifications failed to enrich them in silico.

2. The pull-down products theoretically do not limit to the complexes involved in Okazaki fragment maturation process. In the lagging strand DNA synthesis, there are other important enzymes such as primase/DNA Polymerase alpha, which makes primers, DNA2 which also

participates in primer removal. Their action and dynamic mechanism are what the field is waiting for. However, those complexes were not identified either through mass spectrometry or cryo-EM in the current study. Would synchronizing the cells in S-phase helps to enrich those complexes?

We have used HU treatment to arrest the cells in the DNA synthesis phase (S phase). Under our experimental condition, the cells in S phase have been increased from 50% to 80% upon HU treatment (Rebuttal Figure 1A-1C). During sample preparation, we in fact have tried both conditions (with or without HU treatment). As shown in Rebuttal Figure 1D, SDS-PAGE analysis of the eluted samples after affinity purification showed no significant difference for the two conditions. Please refer to our response to the comment #4 from reviewer 1.

3. This reviewer was not clear if the authors intended to distinguish RNA from DNA to make the claim that RNase H2 binds to double strand DNA to promote FEN1 cleavage. Please clarify it.

We have proposed this hypothesis based on the topology of the DNA structure. Please refer to our response to the comment #15 and #17 from reviewer 1.

4. In addition, the claim that RNase H2 moves from the 5' RNA end to 3' DNA end due to threading through the space between the DNA and DNA Polymerase delta is an imagination and based on the low affinity of DNA Pol delta. Additional evidence is required to support the claim.

The reviewer may have misunderstood what we have intended to propose in the model figure. We did not claim that "RNase H2 moves from the 5' RNA end to 3' DNA end by threading through the space between the DNA and DNA Polymerase delta." Because of the low affinity of Pol delta with DNA, Pol delta could easily dissociate from the DNA during/after primer strand displacement. This would give space for the rebinding of RNaseH2 (which remains associated with PCNA) to the upstream dsDNA region.

5. Other two complexes including the nucleosome class and a two-ringed structure were also identified in the purification fractions 9-12. It was not clear to this reviewer if these two classes both contain PCNA and DNA and if they have anything to do with Okazaki fragment maturation. These two points need to be clarified.

For the nucleosome class, we have no information whether they contain PCNA or not. Since the samples were prepared from PCNA-affinity purification, there was a good chance that at least some of these nucleosome particles were highly flexible PCNA-containing particles. Due to the 2D and 3D averaging, only common features of nucleosomes were resolved in 3D average and 3D reconstruction. For example, histone chaperone such as CAF1, which is abundant in our samples,

is known to directly interact with PCNA and nucleosome/histone tetramers/partially assembled nucleosome (Liu et al, 2023; Orndorff et al, 2024; Shibahara & Stillman, 1999).

For the PCNA dimers, after the three-dimensional classification, we found that there are at least three different forms of PCNA dimers (Rebuttal Figure 4, please also refer to our response to the comment # 7 from reviewer 1). Although the densities of PCNA rings in Classes I and II are of low resolution, it is very clear that these two types of dimers are mediated by other proteins and/or DNA. It is clear that there is DNA in the structure of Class II. In contrast, the dimer structure of Class III is determined at 8.7 Å resolution, which is sufficient for rigid-body docking to determine the polarity of the PCNA ring. However, the Class III dimers appear to be DNA-free, based on the final refined density map. A previous study also reported that there are dimer PCNA rings in mammalian cells, and the two PCNA rings each bind to Pol delta and CAF1, separately (Naryzhny et al, 2005). We acknowledge that the physiological functions of the PCNA dimerization require further exploration.

Referee #3:

The paper by Tian et al reports a cryoEM study of PCNA complexes relevant to Okazaki fragment processing directly from mammalian cells, following transient transfection of HEK293 cells with Strep-tagged PCNA and recovery of PCNA complexes.

FINDING

The main finding of the paper is the recovery and identification of a PCNA-FEN1-RNaseH2-DNA complex, with both FEN1 and RNaseH2 bound via PIP boxes to the PCNA ring. The relative orientation of FEN1 (downstream) and RNaseH2 (upstream) on the DNA leads the authors to postulate a previously unrecognised role of RNaseH2, known to digest the RNA primer, in also assisting the 5'-flap cleaving activity of FEN1. This would be achieved by RNaseH2 binding and stabilisation of the dsDNA directly upstream of FEN1, thus preventing unwanted formation of a 3'-flap after nick translation by pol Delta.

ASSESSMENT

This study adds one more complex to the existing gallery of PCNA-based complexes that are known to be involved in OK fragment maturation, which includes binary and ternary PCNA complexes with Pol delta, FEN1 and Lig1, and the DNA substrate. The isolation and structure determination of the PCNA-FEN1-RNaseH2 complex on DNA is certainly a novel contribution to the field, and the mechanistic interpretation provided by the authors is interesting; however, it remains speculative without any of the follow-up functional experiments that could be thought of to support it (for instance, what is the affinity of RNaseH2 for dsDNA?).

We thank the reviewer for this suggestion. In the revision, we have conducted additional experiments and our results show that:

(1) RNaseH2 has a high affinity to the dsDNA (Rebuttal figure 10).

Our gel shift experiments show that RNaseH2 can bind to dsDNA of different lengths *in vitro*, with the shortest being 5 bp. These results are consistent with a previous study showing that the dissociation equilibrium constant K_d of RNaseH2 with 30 bp ds DNA was 45 nM as measured by fluorescence anisotropy (Coffin *et al*, 2011). The addition of PCNA can significantly improve the dsDNA binding ability of RNaseH2.

(2) The PIP box of RNaseH2A can bind to PCNA *in vitro*

In the PCNA-FEN1-RNaseH2 structure, we find that the PIP box of RNaseH2A binds to PCNA. We confirm that both the PIP box of RNaseH2A and RNaseH2B can bind to PCNA

through size-exclusion chromatography *in vitro* (Rebuttal Figure 6, please also refer to our response to the comment # 13 from reviewer 1). The elution volume of the protein complex also shows that the binding ratio of RNaseH2 to PCNA is roughly 1:1 without DNA. When either of the PIP boxes is mutated, although RNaseH2 still binds to PCNA, the affinity is not as strong as the wild type. This suggests that it is possible that the two PIP boxes of RNaseH2 may simultaneously bind two different subunits of PCNA *in vitro*.

(3) The PIP box of RNaseH2A is not essential for the recruitment of RNaseH2 to the PCNA foci *in vivo*

The PIP box of RNaseH2B has been reported to interact with PCNA and be recruited to the replication fork (Bubeck *et al.*, 2011). To explore the function of RNaseH2A PIP box, we constructed the similar experiment (Rebuttal Figure 7, please also refer to our response to the comment # 13 from reviewer 1). When the PIP box of RNaseH2A mutated, RNaseH2A and RNaseH2B still co-localized with PCNA, indicating that the initial recruitment of RNaseH2 to the replication site does not rely on RNaseH2A PIP box but mainly relies on the interaction between RNaseH2B PIP box and PCNA. The function of the RNaseH2A PIP box remains to be further explored.

We acknowledge that much more experiments are needed to fully test our model. Given the timeframe of the revision and the essence of the current manuscript, we wish to perform in-depth exploration of our model in a future work.

We have also revised the manuscript to emphasize that the model is provocative.

An important issue that puzzles this reviewer is how come the authors could recover from cells and determine the high-resolution structures of FEN1 and FEN1-RNaseH2 bound to PCNA, but failed to recover any of the other complex intermediates that are known to exist and have been characterised before, such as PCNA complexes with Pol Delta and/or FEN1 and/or Lig1? The authors should provide an explanation.

Please also refer to our response to the comment #1 from reviewer 2. Although Pol delta and Lig1 were detected in our purified endogenous complex, we failed to obtain these structures during the data processing probably due to their low abundance and the dynamic nature of the structures containing them.

Reference

Balakrishnan L, Bambara RA (2013) Flap Endonuclease 1. *Annu Rev Biochem* 82: 119-138

Bartsch K, Knittler K, Borowski C, Rudnik S, Damme M, Aden K, Spehlmann ME, Frey N, Saftig P,

Chalaris A *et al* (2017) Absence of RNase H2 triggers generation of immunogenic micronuclei removed by autophagy. *Hum Mol Genet* 26: 3960-3972

Bubeck D, Reijns MA, Graham SC, Astell KR, Jones EY, Jackson AP (2011) PCNA directs type 2 RNase H activity on DNA replication and repair substrates. *Nucleic Acids Res* 39: 3652-3666

Coffin SR, Hollis T, Perrino FW (2011) Functional consequences of the RNase H2A subunit mutations that cause Aicardi-Goutieres syndrome. *J Biol Chem* 286: 16984-16991

Craggs TD, Hutton RD, Brenlla A, White MF, Penedo JC (2014) Single-molecule characterization of Fen1 and Fen1/PCNA complexes acting on flap substrates. *Nucleic Acids Res* 42: 1857-1872

De March M, Merino N, Barrera-Vilarmau S, Crehuet R, Onesti S, Blanco FJ, De Biasio A (2017) Structural basis of human PCNA sliding on DNA. *Nat Commun* 8: 13935

Ghosh D, Kumari S, Raghavan SC (2022) Depletion of RNASEH2 Activity Leads to Accumulation of DNA Double-strand Breaks and Reduced Cellular Survivability in T Cell Leukemia. *J Mol Biol* 434: 167617

Hou Z, Nightingale F, Zhu Y, MacGregor-Chatwin C, Zhang P (2023) Structure of native chromatin fibres revealed by Cryo-ET in situ. *Nat Commun* 14: 6324

Hyjek M, Figiel M, Nowotny M (2019) RNases H: Structure and mechanism. *DNA Repair (Amst)* 84: 102672

Keijzers G, Liu D, Rasmussen LJ (2016) Exonuclease 1 and its versatile roles in DNA repair. *Crit Rev Biochem Mol Biol* 51: 440-451

Liu CP, Yu Z, Xiong J, Hu J, Song A, Ding D, Yu C, Yang N, Wang M, Yu J *et al* (2023) Structural insights into histone binding and nucleosome assembly by chromatin assembly factor-1. *Science* 381: eadd8673

Liu R, Qiu J, Finger LD, Zheng L, Shen B (2006) The DNA-protein interaction modes of FEN-1 with gap substrates and their implication in preventing duplication mutations. *Nucleic Acids Res* 34: 1772-1784

Majka J, Burgers PM (2004) The PCNA-RFC families of DNA clamps and clamp loaders. *Prog Nucleic Acid Res Mol Biol* 78: 227-260

Naryzhny SN, Zhao H, Lee H (2005) Proliferating cell nuclear antigen (PCNA) may function as a double homotrimer complex in the mammalian cell. *J Biol Chem* 280: 13888-13894

Orndorff KS, Veltri EJ, Hoitsma NM, Williams IL, Hall I, Jaworski GE, Majeres GE, Kallepalli S, Vito AF, Struble LR *et al* (2024) Structural Basis for the Interaction Between Yeast Chromatin Assembly Factor 1 and Proliferating Cell Nuclear Antigen. *J Mol Biol* 436: 168695

Oughtred R, Rust J, Chang C, Breitkreutz BJ, Stark C, Willems A, Boucher L, Leung G, Kolas N, Zhang F *et al* (2021) The BioGRID database: A comprehensive biomedical resource of curated protein, genetic, and chemical interactions. *Protein Sci* 30: 187-200

Sartori AA, Lukas C, Coates J, Mistrik M, Fu S, Bartek J, Baer R, Lukas J, Jackson SP (2007) Human CtIP promotes DNA end resection. *Nature* 450: 509-514

Shibahara K, Stillman B (1999) Replication-dependent marking of DNA by PCNA facilitates CAF-1-coupled inheritance of chromatin. *Cell* 96: 575-585

Soman A, Wong SY, Korolev N, Surya W, Lattmann S, Vogirala VK, Chen Q, Berezhnoy NV, van Noort J, Rhodes D *et al* (2022) Columnar structure of human telomeric chromatin. *Nature* 609: 1048-1055

Song F, Chen P, Sun D, Wang M, Dong L, Liang D, Xu RM, Zhu P, Li G (2014) Cryo-EM study of the chromatin fiber reveals a double helix twisted by tetranucleosomal units. *Science* 344: 376-380

Sun H, Ma L, Tsai YF, Abeywardana T, Shen B, Zheng L (2023) Okazaki fragment maturation: DNA flap dynamics for cell proliferation and survival. *Trends Cell Biol* 33: 221-234

Tsutakawa SE, Classen S, Chapados BR, Arvai AS, Finger LD, Guenther G, Tomlinson CG, Thompson P, Sarker AH, Shen B *et al* (2011) Human flap endonuclease structures, DNA double-base flipping, and a unified understanding of the FEN1 superfamily. *Cell* 145: 198-211

Dr. Ning Gao
School of Life Sciences, Peking University
China

22nd Oct 2024

Re: EMBOJ-2024-117698R
Structural insight into the Okazaki fragment maturation by FEN1 and RNaseH2

Dear Ning,

Thank you for submitting your revised manuscript for our consideration. It has now been assessed once more by the original referee 1, whose comments are copied below. As you will see, the referee considers your study significantly improved, but still retains a few specific concerns that would need to be addressed before publication. Most of these points can be addressed with reorganization (pt. 4) and rewriting/down-toning (pts. 2/3/5), but additional (Appendix?) data panels would also be needed to show the still missing control blots (pt.1).

In addition, there are also several editorial issues that would still need to be addressed at this stage:

- As mentioned before: Please make sure to include all relevant funding information not only in the manuscript text, but also in the respective fields in our online submission system.
- Please reduce the number of keywords on the title page to 5 (currently, there are 6), preferring broader terms over particular protein names.
- As we are switching from a free-text author contribution statement towards a more formal statement based on Contributor Role Taxonomy (CRediT) terms, please remove the present Author Contribution section and instead specify each author's contribution(s) directly in the Author Information page of our submission system during upload of the final manuscript. See <https://casrai.org/credit/> for more information.
- Finally, please provide suggestions for a short 'blurb' text prefacing and summing up the study in two sentences (max. 250 characters), followed by 3-5 one-sentence 'bullet points' with brief factual statements of key results of the paper; they will form the basis of an editor-written 'Synopsis' accompanying the online version of the article. Please also upload a synopsis image, which can be used as a "visual title" for the synopsis section of your paper (maybe based on a simplified version of Figure 6, or containing annotated structural snapshots as in Figs. 4/5?). The image should be in PNG or JPG format with the modest dimensions of EXACTLY 550 pixels wide and 300-600 pixels high.

I am therefore returning the study to you once more for revision, to allow you to incorporate these final changes. After that, we should be able to swiftly proceed with formal acceptance and production of the manuscript. Thank you again for this contribution to The EMBO Journal.

Yours sincerely,

Hartmut

- 1) Every manuscript requires a Data Availability section (even if only stating that no deposited datasets are included). Primary datasets or computer code produced in the current study have to be deposited in appropriate public repositories prior to resubmission, and reviewer access details provided in case that public access is not yet allowed. Further information: embopress.org/page/journal/14602075/authorguide#dataavailability
- 2) Each figure legend must specify

9) To facilitate reproducibility and cross-laboratory adoption of methodologies, please structure the Materials & Methods section as outlined in our guide to authors, including a completed Reagents and Tools Table that can be downloaded from our author guidelines as well (<https://www.embopress.org/page/journal/14602075/authorguide#structuredmethods>).

10) Digital image enhancement is acceptable practice, as long as it accurately represents the original data and conforms to community standards. If a figure has been subjected to significant electronic manipulation, this must be clearly noted in the figure legend and/or the 'Materials and Methods' section. The editors reserve the right to request original versions of figures and the original images that were used to assemble the figure. Finally, we generally encourage uploading of numerical as well as gel/blot image source data; for details see: embopress.org/page/journal/14602075/authorguide#sourcedata

At EMBO Press, we ask authors to provide source data for the main manuscript figures. Our source data coordinator will contact you to discuss which figure panels we would need source data for and will also provide you with helpful tips on how to upload and organize the files.

In the interest of ensuring the conceptual advance provided by the work, we recommend submitting a revision within 3 months (20th Jan 2025). Please discuss the revision progress ahead of this time with the editor if you require more time to complete the revisions. Use the link below to submit your revision:

Link Not Available

Referee #1:

This resubmission from Tian et al shows substantial improvements of the manuscript. Besides text revisions, the authors have included new experimental data for the role of RnaseH2A's PIP box in localizing RNaseH2 to PCNA and confirming DNA binding by RNase H2. However several issues remain a concern and need to be addressed.

1) The authors provide no experimental data showing the extend of overexpression of PCNA in cells used for purification of

PCNA complexes. This should be included so that readers can come to their own conclusions whether the extent of overexpression may be concerning or not.

2) The claim of PCNA dimerization remains highly speculative. In several class averages, there appears no density between what the authors argue are PCNA rings. Without higher resolution structures, experimental support of physiological significance, or identification of the bridging proteins, this data does not add but distracts from the main conclusions of the paper.

3) The conclusion that the cryoEM samples contain di- and trinucleosomes with similarities to the 30 nm fiber remains unconvincing. This statement is not relevant for the proposed mechanism of Okazaki fragment processing and should be removed.

4) The authors introduce new experimental data in the discussion. This should be moved to the results section.

5) Referring to a K_d of 35 nM for Pol delta binding to DNA as extremely low binding affinity is confusing and misleading. Most researchers would consider a K_d of 35 nM high affinity.

Comments for the revision

Referee #1:

This resubmission from Tian et al shows substantial improvements of the manuscript. Besides text revisions, the authors have included new experimental data for the role of RnaseH2A's PIP box in localizing RNaseH2 to PCNA and confirming DNA binding by RNase H2. However, several issues remain a concern and need to be addressed.

1) The authors provide no experimental data showing the extend of overexpression of PCNA in cells used for purification of PCNA complexes. This should be included so that readers can come to their own conclusions whether the extend of overexpression may be concerning or not.

Thanks for the reviewer's suggestion. We have added a western blot figure in the Appendix showing the extent of PCNA overexpression. Although the overexpression level of PCNA was high through transient plasmid transfection, we found that twin-strep-PCNA were efficiently incorporated into the chromatin fractions (Appendix Fig. S1).

2) The claim of PCNA dimerization remains highly speculative. In several class averages, there appears no density between what the authors argue are PCNA rings. Without higher resolution structures, experimental support of physiological significance, or identification of the bridging proteins, this data does not add but distracts from the main conclusions of the paper.

Most of the text on the PCNA dimers in the results and discussion has now been removed.

3) The conclusion that the cryoEM samples contain di- and trinucleosomes with similarities to the 30 nm fiber remains unconvincing. This statement is not relevant for the proposed mechanism of Okazaki fragment processing and should be removed.

Suggestion is taken. The description of the di- and tri-nucleosome has been removed in the revision.

4) The authors introduce new experimental data in the discussion. This should be moved to the results section.

We have moved the new experimental data about PIP box of RNaseH2A to the results section.

5) Referring to a Kd of 35 nM for Pol delta binding to DNA as extremely low binding affinity is confusing and misleading. Most researchers would consider a Kd of 35 nM high affinity.

Compared with the Kd of RNaseH2 binding to dsDNA, the Kd of Pol delta binding to DNA is relatively low. The manuscript has been revised as follows: "In stark contrast, the DNA binding affinity of mammalian Pol delta is 70-fold lower".

Dr. Ning Gao
School of Life Sciences, Peking University
Yiheyuan Road 5#
Beijing, Beijing 100871
China

28th Oct 2024

Re: EMBOJ-2024-117698R1
Structural insight into Okazaki fragment maturation mediated by PCNA-bound FEN1 and RNaseH2

Dear Dr. Gao,

Thank you for submitting your final revised manuscript for our consideration. I am pleased to inform you that we have now accepted it for publication in The EMBO Journal.

Yours sincerely,

Hartmut Vodermaier
